# Provably Efficient Policy-Reward Co-Pretraining for Adversarial Imitation Learning

Tian Xu [1]  Zexuan Chen [1]  Zhilong Zhang [1]  Yi-Chen Li [1]  Chenyang Wang [1]  Lei Yuan [1]  Yang Yu [1]

## Abstract

Adversarial imitation learning (AIL) achieves high-quality imitation compared to behavioral cloning (BC), but demands substantial online environment interaction. Recent empirical work has explored initializing AIL algorithms with BC-pretrained policies to address this limitation, yet a rigorous theoretical understanding of pretraining's role in AIL remains elusive. This paper provides a systematic theoretical analysis and introduces principled pretraining algorithms for accelerating AIL. We begin by analyzing AIL with policy pretraining alone, identifying reward error as the dominant source of suboptimality. This reveals a critical and previously overlooked gap: the absence of reward pretraining. Motivated by this finding, we develop a principled policy–reward co-pretraining approach grounded in a reward-shaping analysis. Our analysis uncovers a fundamental connection between expert policies and shaping rewards, which naturally gives rise to CoPT-AIL, an approach that jointly pretrains both policy and reward through a single BC procedure. We prove that CoPT-AIL achieves an improved imitation gap bound over standard AIL, establishing the first theoretical guarantee for the benefits of pretraining in AIL. Experimental results confirm CoPT-AIL's superior performance over existing AIL methods.

## 1. Introduction

Imitation learning (IL) (Argall et al., 2009; Osa et al., 2018) is a foundational technique in artificial intelligence that enables agents to acquire complex behaviors by mimicking expert demonstrations. It has achieved remarkable success across diverse domains, including autonomous driving (Pan et al., 2017), generalist robot learning (Brohan et al., 2023; Mees et al., 2024), and language modeling (Brown et al., 2020).

IL comprises two primary methodological families: behavioral cloning (BC) and adversarial imitation learning (AIL). BC is an offline approach that directly applies supervised learning to expert demonstrations (Pomerleau, 1991; Ross et al., 2011; Brantley et al., 2020). While conceptually simple, BC is vulnerable to compounding errors (Syed & Schapire, 2010), which degrade imitation quality. AIL (Abbeel & Ng, 2004; Syed & Schapire, 2007; Ho & Ermon, 2016; Kostrikov et al., 2019), by contrast, seeks to match the expert's state-action distribution via a minimax optimization framework. It alternates between recovering an adversarial reward function that maximizes the policy value gap between expert and learner, and updating the policy to minimize this gap. Because this optimization requires online environment interaction, AIL is classified as an online method. Both theoretical analysis (Rajaraman et al., 2020; Xu et al., 2020; 2026a) and empirical evidence (Ho & Ermon, 2016; Kostrikov et al., 2019; Ghasemipour et al., 2019) confirm that AIL effectively overcomes BC's compounding error problem and achieves high-quality imitation.

While AIL demonstrates superior performance, its reliance on extensive online environment interactions presents a significant limitation (Ho & Ermon, 2016). To mitigate this limitation, researchers have explored various approaches to combine AIL with BC (Jena et al., 2021; Orsini et al., 2021; Haldar et al., 2023; Watson et al., 2023; Yue et al., 2024). The most intuitive approach involves pretraining policies using BC, then fine-tuning them with AIL through online interactions (Ho & Ermon, 2016). However, empirical studies consistently show that this strategy provides minimal benefits (Sasaki et al., 2018; Jena et al., 2021; Orsini et al., 2021; Yue et al., 2024). The pretrained policy's performance typically degrades during early AIL training, negating most advantages from the initial BC phase.

Recent work has proposed incorporating reward pretraining to overcome this limitation (Watson et al., 2023; Yue et al., 2024). Specifically, Watson et al. (2023) pretrains a reward under which the BC policy becomes optimal, while

[1] National Key Laboratory for Novel Software Technology and School of Artificial Intelligence, Nanjing University, China. Correspondence to: Yang Yu <yuy@nju.edu.cn>.

*Proceedings of the 43rd International Conference on Machine Learning*, Seoul, South Korea. PMLR 306, 2026. Copyright 2026 by the author(s).

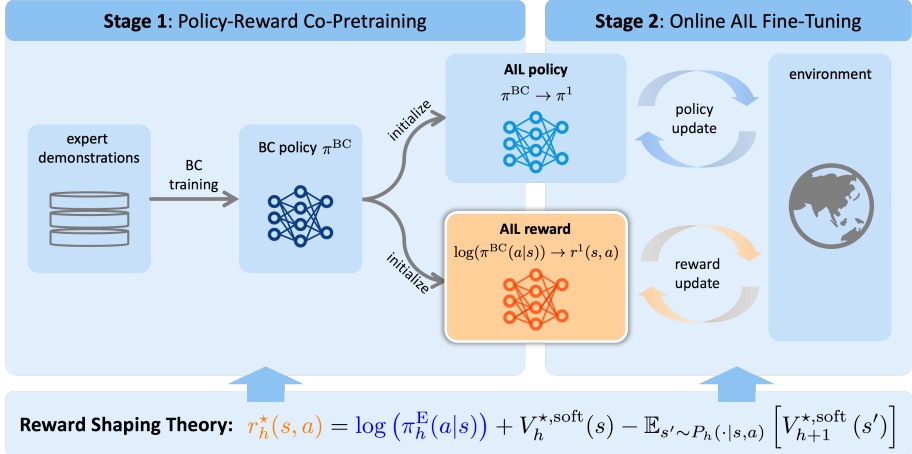

*Figure 1.* Illustration of CoPT-AIL. **Stage 1 (Policy-Reward Co-Pretraining):** A BC policy $\pi^{\mathrm{BC}}$ is trained on expert demonstrations and used to jointly initialize both the AIL policy and reward function. **Stage 2 (Online AIL Fine-Tuning):** The co-pretrained policy and reward are fine-tuned through standard online AIL via environment interaction.

Yue et al. (2024) builds on the closed-form solution of the GAIL reward function (Ho & Ermon, 2016) and leverages supplementary datasets to approximate it. Despite empirical gains in certain settings, the theoretical role of reward pretraining in AIL remains poorly understood, which risks limiting future algorithmic progress.

This paper aims to bridge the gap between theory and practice by providing rigorous theoretical guarantees on the *imitation gap* (i.e., performance difference between the expert and learner) and developing effective pre-training algorithms to accelerate AIL. Our key contributions are threefold.

- First, we develop a theoretical analysis for AIL with policy pretraining alone, identifying reward error as the dominant error source and establishing a sharp theoretical baseline. Specifically, the overall imitation gap can be decomposed into policy error and reward error. We prove that while policy pre-training effectively reduces the cumulative policy error, the specific reward error induced by random reward initialization remains a persistent bottleneck (Proposition 1). This rigorous diagnosis serves a dual purpose: it mathematically isolates the reward error as the precise target for our algorithmic design, and establishes a tight theoretical baseline for the subsequent theoretical comparison.

- Guided by this theoretical diagnosis, we derive a principled policy–reward co-pretraining method, CoPT-AIL, grounded in a reward-shaping analysis. We prove that inferring a shaping reward, rather than the original true reward, is already sufficient to reduce reward error, thereby circumventing the reward ambiguity issue (Proposition 2). Crucially, our analysis reveals a fundamental connection between the expert policy and

shaping reward, naturally giving rise to the approach of jointly pretraining policies and rewards through a single BC procedure. This yields our complete algorithm CoPT-AIL, **AIL** with Policy-Reward **Co-Pre**training, which is illustrated in Figure 1.

- Finally, we provide a rigorous theoretical analysis demonstrating CoPT-AIL's superiority over prior AIL approaches. Our theoretical results show that CoPT-AIL can provably reduce reward error through reward pretraining, achieving an improved imitation gap bound compared to standard AIL without pretraining under mild assumptions (Theorem 1). To the best of our knowledge, this represents the first theoretical guarantee for the efficiency gains of pretraining in AIL. Experimental evaluation confirms CoPT-AIL's superior performance over existing methods.

## 2. Preliminaries

**Markov Decision Process.** We consider episodic Markov Decision Processes (MDPs) represented by the tuple $\mathcal{M} = (\mathcal{S}, \mathcal{A}, P, r^\star, H, \rho)$, where $\mathcal{S}$ and $\mathcal{A}$ denote the state and action spaces, respectively, $H$ is the planning horizon, and $\rho$ is the initial state distribution. The transition dynamics is characterized by $P = \{P_1, \ldots, P_H\}$, where $P_h(s_{h+1}|s_h, a_h)$ gives the probability of transitioning to state $s_{h+1}$ from state $s_h$ upon taking action $a_h$ at step $h \in [H]$. The reward is characterized by $r^\star = \{r_1^\star, \ldots, r_H^\star\}$, where without loss of generality, $r_h^\star : \mathcal{S} \times \mathcal{A} \to [0, 1]$ for all $h \in [H]$.

A policy $\pi = \{\pi_1, \ldots, \pi_H\}$ maps states to action distributions, with $\pi_h : \mathcal{S} \to \Delta(\mathcal{A})$, where $\Delta(\mathcal{A})$ denotes the probability simplex over actions. Here, $\pi_h(a|s)$ represents the probability of selecting action $a$ in state $s$ at step $h$. The interaction protocol proceeds as follows: each episode be-

gins with the environment sampling an initial state $s_1 \sim \rho$. At each step $h$, the agent observes state $s_h$, selects action $a_h \sim \pi_h(\cdot|s_h)$, receives reward $r_h^\star(s_h, a_h)$, and transitions to the next state $s_{h+1} \sim P_h(\cdot|s_h, a_h)$. The episode terminates after $H$ steps. We evaluate policy performance using the expected cumulative reward:

$$V^\pi := \mathbb{E}\left[\sum_{h=1}^{H} r_h^\star(s_h, a_h) \middle| a_h \sim \pi_h(\cdot|s_h),\right.$$
$$\left. s_{h+1} \sim P_h(\cdot|s_h, a_h), \forall h \in [H]\right].$$

The Q-function is defined as $Q_h^\pi(s, a) := \mathbb{E}\left[\sum_{h'=h}^{H} r_{h'}^\star(s_{h'}, a_{h'}) \middle| (s_h, a_h) = (s, a), \pi\right]$. We also define the state visitation distribution $d_h^\pi(s) := \mathbb{P}^\pi(s_h = s)$ and state-action visitation distribution $d_h^\pi(s, a) := \mathbb{P}^\pi(s_h = s, a_h = a)$.

**Imitation Learning.** The goal of imitation learning (IL) is to acquire a high-quality policy *without* access to the reward function $r^\star$. To achieve this, we assume the learner has access to an expert dataset consisting of $N$ trajectories collected by the expert policy $\pi^E$:

$$\mathcal{D}^E = \{\tau^i = (s_1^i, a_1^i, s_2^i, a_2^i, \ldots, s_H^i, a_H^i);$$
$$a_h^i \sim \pi_h^E(\cdot|s_h^i), s_{h+1}^i \sim P_h(\cdot|s_h^i, a_h^i), \forall h \in [H]\}_{i=1}^N,$$

The learner uses this dataset $\mathcal{D}^E$ to learn a policy that mimics the expert's behavior. We measure imitation quality using the *imitation gap* (Abbeel & Ng, 2004; Ross & Bagnell, 2010; Rajaraman et al., 2020), defined as $V^{\pi^E} - V^\pi$, where $\pi$ is the learned policy. Essentially, we hope that the learned policy can perfectly mimic the expert such that the imitation gap is small.

Typical IL works (Ng & Russell, 2000; Abbeel & Ng, 2004) often assume that the expert policy is optimal with respect to the true reward $r^\star$, which suffers from the issue that degenerate constant rewards can induce the same expert policy (Ziebart et al., 2008). Following maximum entropy inverse reinforcement learning (Ziebart et al., 2008; Bloem & Bambos, 2014), we avoid this issue by assuming that the expert is a soft-optimal policy (Haarnoja et al., 2018; Geist et al., 2019) with respect to $r^\star$. Formally, the expert policy is given by

$$\pi_h^E(a|s) = \exp\left(Q_h^{\star,\text{soft}}(s, a) - V_h^{\star,\text{soft}}(s)\right). \quad (1)$$

Here $Q_h^{\star,\text{soft}}(s, a)$ and $V_h^{\star,\text{soft}}(s)$ denote the soft-optimal Q-function and value function, respectively.

**Behavioral Cloning.** As a classical IL method, behavioral cloning (BC) (Pomerleau, 1991) performs maximum likelihood estimation (MLE) to mimic the expert.

$$\pi^{\text{BC}} = \underset{\pi \in \Pi}{\arg\max} \sum_{i=1}^{N} \sum_{h=1}^{H} \log\left(\pi_h(a_h^i|s_h^i)\right). \quad (2)$$

Here $\Pi$ is the set of all policies. This optimization problem can be solved entirely using pre-collected expert data without any environment interaction, making BC a purely offline method. However, this offline nature introduces a fundamental limitation: BC is susceptible to compounding errors (Ross & Bagnell, 2010), resulting in poor imitation performance given limited demonstrations.

**Adversarial Imitation Learning.** As another prominent class of IL methods, adversarial imitation learning (AIL) imitates expert behavior through a game-theoretic approach.

$$\max_{\pi \in \Pi} \min_{r \in \mathcal{R}} V_r^\pi - V_r^{\pi^E}. \quad (3)$$

Here $V_r^\pi$ denotes the value of policy $\pi$ under reward $r$ and $\mathcal{R} := \{r : \forall (s, a, h) \in \mathcal{S} \times \mathcal{A} \times [H], r_h(s, a) \in [0, 1]\}$ denotes the reward class. In this minimax objective, AIL infers a reward function that maximizes the value gap between the expert policy and the learner's policy. Subsequently, it learns a policy that minimizes this value gap using the inferred reward. Note that the outer optimization problem concerning the policy is equivalent to a reinforcement learning (RL) problem under the inferred reward $r$. Solving RL problems typically requires online environment interactions, marking AIL as an online approach. AIL has proven to mitigate the compounding-error problem in BC both theoretically (Xu et al., 2020; Rajaraman et al., 2020; Xu et al., 2026a) and empirically (Ho & Ermon, 2016; Kostrikov et al., 2019; Ghasemipour et al., 2019), achieving good imitation performance. However, AIL relies on extensive online environment interactions, presenting a significant limitation in scenarios where such interactions are expensive.

## 3. The Critical Role of Reward Pretraining in Adversarial Imitation Learning

A natural approach to improve the interaction efficiency of AIL involves first pretraining policies via BC, then fine-tuning them through AIL with online interactions (Ho & Ermon, 2016). This intuitive strategy leverages BC to establish an acceptable initial policy before engaging in interaction-expensive adversarial learning. However, numerous empirical works (Sasaki et al., 2018; Jena et al., 2021; Orsini et al., 2021; Yue et al., 2024) have consistently found that policy pretraining alone provides minimal benefits. In particular, these works observed that the policy quality deteriorates rapidly at the beginning of AIL training, negating

most advantages gained from the initial BC phase. This phenomenon suggests fundamental limitations of AIL with policy pretraining alone, yet a rigorous theoretical explanation of why these limitations arise remains elusive.

To address this theoretical gap, we develop a rigorous analysis for AIL with policy pretraining. Concretely, we formally examine a standard AIL procedure with BC-pretrained policies, outlined in Algorithm 1.

---

**Algorithm 1** Adversarial Imitation Learning with Policy Pretraining Alone

---

**Input:** Randomly initialized reward $r^1$ and demonstrations $\mathcal{D}^{\mathrm{E}}$.

1: Pretrain a policy via BC based on Eq.(2): $\pi^1 \leftarrow \pi^{\mathrm{BC}}$.
2: **for** $k = 1, 2, \ldots, K-1$ **do**
3:      Calculate the Q-value function $\{Q_h^{\pi^k, r^k}\}_{h=1}^H$ for policy $\pi^k$.
4:      Update the policy by KL-regularized policy optimization:
$$\pi_h^{k+1}(\cdot|s) = \operatorname*{argmax}_{p \in \Delta(\mathcal{A})} \left\{ \mathbb{E}_{a \sim p(\cdot)}[Q_h^{\pi^k, r^k}(s,a)] - \frac{1}{\eta} D_{\mathrm{KL}}(p(\cdot), \pi_h^k(\cdot|s)) \right\}.$$
5:      Update the reward by solving the optimization problem:
$$r^{k+1} = \operatorname*{argmin}_{r \in \mathcal{R}} \left\{ \mathbb{E}_{\tau \sim \pi^{k+1}} \left[ \sum_{h=1}^H r_h(s_h, a_h) \right] - \mathbb{E}_{\tau \sim \mathcal{D}^{\mathrm{E}}} \left[ \sum_{h=1}^H r_h(s_h, a_h) \right] \right\}.$$
6: **end for**
**Output:** $\bar{\pi}$ sampled uniformly from $\{\pi^1, \ldots, \pi^K\}$.

---

Algorithm 1 operates in two stages. First, we pretrain policies through BC on expert demonstrations. Second, we conduct the online AIL process, which alternates between policy and reward updates. During policy updates, we employ KL-regularized policy optimization (Shani et al., 2020; Cai et al., 2020) to solve the outer RL problem in Eq.(3). During reward updates, with the newly recovered policy $\pi^{k+1}$, we update the reward by minimizing the policy value difference between $\pi^{k+1}$ and $\pi^{\mathrm{E}}$, i.e., $\min_{r \in \mathcal{R}} V_r^{\pi^{k+1}} - \widehat{V}_r^{\pi^{\mathrm{E}}}$, where $\widehat{V}_r^{\pi^{\mathrm{E}}} := \mathbb{E}_{\tau \sim \mathcal{D}^{\mathrm{E}}}[\sum_{h=1}^H r_h(s_h, a_h)]$ represents an empirical estimation of $V_r^{\pi^{\mathrm{E}}}$ based on demonstrations. Finally, following the standard online-to-batch conversion technique (Orabona, 2019), Algorithm 1 outputs a policy uniformly sampled from the recovered policies throughout training.

The following proposition provides the imitation gap bound of AIL with policy pretraining alone.

**Proposition 1.** *Consider adversarial imitation learning with policy pretraining shown in Algorithm 1. For any*

$\delta \in (0, 1)$*, with probability at least $1 - \delta$, it holds that*

$$V^{\pi^{\mathrm{E}}} - V^{\bar{\pi}} \leq \underbrace{\frac{1}{K}\left(V_{r^\star}^{\pi^{\mathrm{E}}} - V_{r^\star}^{\pi^1} - \left(V_{r^1}^{\pi^{\mathrm{E}}} - V_{r^1}^{\pi^1}\right)\right)}_{\text{reward error}}$$
$$+ 2\sqrt{\frac{2|\mathcal{S}||\mathcal{A}|H^2 \log(H/\delta)}{N}}$$
$$+ \underbrace{\frac{1}{\eta K}\mathbb{E}\left[\sum_{h=1}^H D_{\mathrm{KL}}(\pi_h^{\mathrm{E}}(\cdot|s_h), \pi_h^1(\cdot|s_h))\Big|\pi^{\mathrm{E}}\right]}_{\text{policy error}}$$
$$+ \frac{\eta}{2}H^3. \tag{4}$$

*Furthermore, consider pretraining policies via BC (i.e., $\pi^1 := \pi^{\mathrm{BC}}$) and choose stepsize $\eta = \widetilde{\Theta}\left(\sqrt{(|\mathcal{S}||\mathcal{A}|)/(H^2 K N)}\right)$, we have that*

$$V^{\pi^{\mathrm{E}}} - V^{\bar{\pi}} \precsim \frac{1}{K}\left(V_{r^\star}^{\pi^{\mathrm{E}}} - V_{r^\star}^{\pi^1} - \left(V_{r^1}^{\pi^{\mathrm{E}}} - V_{r^1}^{\pi^1}\right)\right)$$
$$+ \sqrt{\frac{|\mathcal{S}||\mathcal{A}|H^2 \log(H/\delta)}{N}}$$
$$+ \sqrt{\frac{|\mathcal{S}||\mathcal{A}|H^4 \log^2(HN^2/\delta)}{KN}}. \tag{5}$$

The complete proof is provided in Appendix A.1. Eq.(4) in Proposition 1 reveals that the imitation gap of AIL with policy pretraining contains two fundamental error components: reward error and policy error. The reward error quantifies the discrepancy between the true reward $r^\star$ and the initial reward $r^1$ through value difference. The policy error specifically measures the KL divergence between the expert policy $\pi^{\mathrm{E}}$ and the initial policy $\pi^1$. Additionally, the second term in the RHS of Eq.(4) captures the statistical error arising from the finite number of expert demonstrations, while the last term represents the optimization error that occurs in performing KL-regularized policy updates.

Policy pretraining via BC (i.e., $\pi^1 \leftarrow \pi^{\mathrm{BC}}$) effectively reduces the policy error, as the BC policy approximates the expert policy far better than a random initialization. Formally, applying the theoretical guarantee of BC (Tiapkin et al., 2024) to bound the policy error yields the sharper result in Eq.(5). However, a critical limitation persists: the reward error remains large because $r^1$ is randomly initialized and may therefore be arbitrarily far from $r^\star$. This reward error can substantially inflate the overall imitation gap, particularly in the early stages of training when $K$ is small. This result thus offers a formal theoretical explanation for why AIL with policy pretraining alone yields minimal performance gains: while BC pretraining effectively reduces the policy error, the reward error, left entirely unaddressed, persists as the dominant bottleneck in the imitation gap.

Beyond the explanatory value, Proposition 1 prescribes a clear target: to achieve meaningful gains over standard AIL, one must reduce the reward error. This insight directly motivates our algorithm introduced in the next section.

# 4. Policy-Reward Co-Pretraining for Adversarial Imitation Learning

Building on the theoretical insights from the previous section, we propose a joint pretraining approach for both policies and rewards to accelerate AIL. We first introduce a principled method for reward pretraining, then provide a rigorous theoretical analysis demonstrating its effectiveness in reducing the imitation gap.

## 4.1. Method

Building on Proposition 1, we develop a reward pretraining method to reduce the reward error represented by the term $(V_{r^\star}^{\pi^E} - V_{r^\star}^\pi) - (V_r^{\pi^E} - V_r^\pi)$. We refer to this term as the *relative policy evaluation error*, as it quantifies the discrepancy in evaluating the relative value difference between policies $\pi^E$ and $\pi$. Based on the well-known simulation lemma (Kearns & Singh, 2002), a natural approach to reducing this error would be to pretrain a reward $r$ that closely approximates the original true reward $r^\star$, ensuring $|r_h^\star(s, a) - r_h(s, a)|$ is small. However, the reward ambiguity issue fundamentally prevents recovering a reward function close to $r^\star$, even with complete knowledge of the expert policy and MDP (Cao et al., 2021; Metelli et al., 2021; Rolland et al., 2022).

To circumvent this limitation, we argue that learning a reward close to the original $r^\star$ is not necessary for reducing the relative policy evaluation error. Instead, we demonstrate that learning an accurate *shaping reward* (Ng et al., 1999) is already sufficient. We begin by introducing the formal definition of the shaping reward [1].

**Definition 1** (Shaping Rewards (Ng et al., 1999))**.** *In an episodic MDP, for a reward function $r$ and potential shaping functions $\{\Phi_h : \mathcal{S} \to \mathbb{R}\}_{h=1}^{H+1}$ with $\Phi_{H+1} \equiv 0$, the shaping reward is defined as:*

$$\widetilde{r}_h(s, a) := r_h(s, a) - \Phi_h(s) + \mathbb{E}_{s' \sim P_h(\cdot|s,a)}[\Phi_{h+1}(s')],$$

*for all $(s, a, h) \in \mathcal{S} \times \mathcal{A} \times [H]$.*

The core design principle in shaping rewards involves a telescoping structure within the shaping functions, which theoretically guarantees that reward shaping preserves optimal policies (Ng et al., 1999).

---

[1] Ng et al. (1999) originally proposed shaping rewards for infinite-horizon discounted MDPs; we present the episodic adaptation here.

Crucially, the following proposition shows that value differences $V^{\pi'} - V^\pi$ remain identical under both the original reward $r$ and its corresponding shaping reward $\widetilde{r}$.

**Proposition 2.** *For any pair of policies $\pi$ and $\pi'$, consider an arbitrary reward $r$ and its shaping reward $\widetilde{r}$ defined by $\widetilde{r}_h(s, a) := r_h(s, a) - \Phi_h(s) + \mathbb{E}_{s' \sim P_h(\cdot|s,a)}[\Phi_{h+1}(s')]$ with potential-based shaping functions $\{\Phi_h\}_{h=1}^{H+1}$, it holds that*

$$V_r^{\pi'} - V_r^\pi = V_{\widetilde{r}}^{\pi'} - V_{\widetilde{r}}^\pi.$$

The insight is that while individual policy values may differ between the original and shaping rewards, their relative differences remain invariant. Intuitively, according to the telescoping argument, the policy values of the original reward and the shaping reward only differ in the shaping value at the initial state, which cancels out when computing value differences. Proposition 2 has an important implication for our reward pretraining approach. It establishes that

$$(V_{r^\star}^{\pi^E} - V_{r^\star}^\pi) - (V_r^{\pi^E} - V_r^\pi)$$
$$= (V_{\widetilde{r}^\star}^{\pi^E} - V_{\widetilde{r}^\star}^\pi) - (V_r^{\pi^E} - V_r^\pi),$$

where $\widetilde{r^\star}$ is certain shaping reward of $r^\star$. This reveals that learning a reward function close to any *shaping* reward $\widetilde{r}^\star$ is sufficient for reducing the relative policy evaluation error. As such, we do not need to recover the original reward $r^\star$ itself.

Having established that learning an accurate shaping reward is sufficient for reducing the reward error, we now develop a principled method to infer such a reward. Our key insight is that the log-probability of the expert policy is exactly a shaping reward. To see this, we take the logarithm of Eq.(1) and obtain

$$\log(\pi_h^E(a|s)) = Q_h^{\star,\text{soft}}(s, a) - V_h^{\star,\text{soft}}(s). \quad (6)$$

Furthermore, the soft Bellman equation gives

$$Q_h^{*,\text{soft}}(s, a) = r_h^*(s, a) + \mathbb{E}_{s' \sim P_h(\cdot|s,a)}\left[V_{h+1}^{*,\text{soft}}(s')\right].$$

Plugging the above equation into Eq.(6) yields the following characterization.

$$\log(\pi_h^E(a|s)) = r_h^\star(s, a) - V_h^{\star,\text{soft}}(s)$$
$$+ \mathbb{E}_{s' \sim P_h(\cdot|s,a)}[V_{h+1}^{\star,\text{soft}}(s')].$$

Crucially, we observe that $\widetilde{r}_h^\star(s, a) = \log(\pi_h^E(a|s))$ is exactly a shaping reward of $r_h^\star(s, a)$ with respect to the potential-based shaping functions $\{V_h^{\star,\text{soft}}\}_{h=1}^{H+1}$. This shaping reward has an intuitive interpretation: it assigns greater values to actions with higher probabilities under the expert.

This reward shaping characterization naturally motivates our reward pretraining approach. We first learn a BC policy $\pi^{\mathrm{BC}}$, then initialize the reward as $r_h^1(s, a) = \log \pi_h^{\mathrm{BC}}(a|s)$, reusing the BC log-likelihood as a separate reward signal. Since $\pi_h^{\mathrm{BC}}(a|s)$ approximates $\pi_h^{\mathrm{E}}(a|s)$ via maximum likelihood estimation, the pretrained reward $r_h^1(s, a)$ is close to the target shaping reward $\widetilde{r}_h^\star(s, a)$.

It is worth noting that while $\pi^{\mathrm{BC}}$ suffers from compounding errors during long-horizon rollouts, this does not meaningfully degrade the quality of the pretrained reward. Compounding errors are a phenomenon of *execution*: they arise during sequential, auto-regressive rollouts due to covariate shift, where small mistakes accumulate over time. The shaping reward $r_h^1(s, a) = \log \pi_h^{\mathrm{BC}}(a|s)$, by contrast, is a matter of *evaluation*—it is assessed locally on individual state-action pairs, entirely bypassing sequential rollout. Since $\pi^{\mathrm{BC}}$ is trained via maximum likelihood, its single-step predictions $\log \pi_h^{\mathrm{BC}}(a|s)$ remain accurate near the expert distribution, yielding a dense and informative signal that steers the agent toward high-probability expert regions. This provides an effective warm start for the online AIL process, which then leverages online interactions to correct any residual errors. Combining AIL with this joint pretraining of policies and rewards yields our overall algorithm, **AIL** with Policy-Reward **Co-Pre**training (**CoPT-AIL**), outlined in Algorithm 2. The key step that distinguishes CoPT-AIL from standard AIL with policy pretraining alone is highlighted in green.

At a high level, the reward-shaping-based analysis reveals a fundamental connection between the expert policy and shaping reward, enabling a unified approach to policy and reward pretraining. This integration allows us to derive both components from a single learning procedure, eliminating the need for a separate reward learning step. The resulting computational efficiency gains are particularly valuable when working with large-parameter models. Moreover, this connection suggests parameterizing rewards with policy models, which could be of independent interest.

### 4.2. Theoretical Analysis

We now provide a rigorous theoretical analysis establishing the superiority of CoPT-AIL.

**Theorem 1.** *Consider adversarial imitation learning with policy-reward co-pretraining (Algorithm 2). For any fixed $\delta \in (0, 1)$, with probability at least $1 - \delta$, the reward error satisfies*

$$
\frac{1}{K}\left(V_{r^\star}^{\pi^{\mathrm{E}}} - V_{r^\star}^{\pi^1} - \left(V_{r^1}^{\pi^{\mathrm{E}}} - V_{r^1}^{\pi^1}\right)\right)
$$
$$
\precsim \frac{C|\mathcal{S}||\mathcal{A}|H^2 \log^2\left(|\mathcal{S}||\mathcal{A}|HN^2/\delta\right)}{KN}, \tag{7}
$$

*where $C := \max_{(s,h)\in\mathcal{S}\times[H]} d_h^{\pi^{\mathrm{BC}}}(s)/d_h^{\pi^{\mathrm{E}}}(s)$. Further-*

---

**Algorithm 2** Adversarial Imitation Learning with Policy-Reward Co-Pretraining

**Input:** Demonstrations $\mathcal{D}^{\mathrm{E}}$.
1: Pretrain a policy via BC based on Eq.(2): $\pi^1 \leftarrow \pi^{\mathrm{BC}}$.
2: Copy the BC policy as an independent reward:
   $r_h^1(s, a) = \log(\pi_h^{\mathrm{BC}}(a|s))$.
3: **for** $k = 1, 2, \ldots, K - 1$ **do**
4:    Calculate Q-value function $\{Q_h^{\pi^k, r^k}\}_{h=1}^H$ for policy $\pi^k$.
5:    Update the policy by KL-regularized policy optimization:
$$
\pi_h^{k+1}(\cdot|s) = \underset{p\in\Delta(\mathcal{A})}{\operatorname{argmax}}\left\{\mathbb{E}_{a\sim p(\cdot)}[Q_h^{\pi^k, r^k}(s, a)]\right.
$$
$$
\left. - \frac{1}{\eta}D_{\mathrm{KL}}(p(\cdot), \pi_h^k(\cdot|s))\right\}.
$$
6:    Update the reward by solving the optimization:
$$
r^{k+1} = \underset{r\in\mathcal{R}}{\operatorname{argmin}}\left\{\mathbb{E}_{\tau\sim\pi^{k+1}}\left[\sum_{h=1}^H r_h(s_h, a_h)\right]\right.
$$
$$
\left. - \mathbb{E}_{\tau\sim\mathcal{D}^{\mathrm{E}}}\left[\sum_{h=1}^H r_h(s_h, a_h)\right]\right\}.
$$
7: **end for**
**Output:** $\bar{\pi}$ sampled uniformly from $\{\pi^1, \ldots, \pi^K\}$.

---

*more, the imitation gap satisfies*

$$
V^{\pi^{\mathrm{E}}} - V^{\bar{\pi}} \precsim \frac{C|\mathcal{S}||\mathcal{A}|H^2 \log^2\left(|\mathcal{S}||\mathcal{A}|HN^2/\delta\right)}{KN}
$$
$$
+ \sqrt{\frac{|\mathcal{S}||\mathcal{A}|H^2 \log(H/\delta)}{N}} \tag{8}
$$
$$
+ \sqrt{\frac{|\mathcal{S}||\mathcal{A}|H^4 \log^2(HN^2/\delta)}{KN}}.
$$

The complete proof is deferred to Appendix A.3. Theorem 1 shows that reward pretraining reduces the reward error to $\widetilde{\mathcal{O}}(C|\mathcal{S}||\mathcal{A}|H^2/(KN))$, which decays rapidly with the number of expert trajectories $N$. This confirms that our approach effectively leverages expert demonstrations to obtain a high-quality initial reward.

Turning to the overall imitation gap, CoPT-AIL achieves a bound of $\widetilde{\mathcal{O}}((C|\mathcal{S}||\mathcal{A}|H^2)/(KN) + \sqrt{|\mathcal{S}||\mathcal{A}|H^2/N} + \sqrt{|\mathcal{S}||\mathcal{A}|H^4/(KN)})$. By comparison, Shani et al. (2021) established that standard AIL without pretraining achieves $\widetilde{\mathcal{O}}(\sqrt{|\mathcal{S}||\mathcal{A}|H^2/K} + \sqrt{|\mathcal{S}||\mathcal{A}|H^3/N} + \sqrt{H^4/K})$. CoPT-AIL therefore yields a strictly better bound whenever the number of expert trajectories satisfies $N \gtrsim C\sqrt{|\mathcal{S}||\mathcal{A}|H^2/K^2}$. Intuitively, when sufficiently many demonstrations are available, jointly pretraining both the

---

[2]A detailed term-by-term comparison is provided in Appendix A.4.

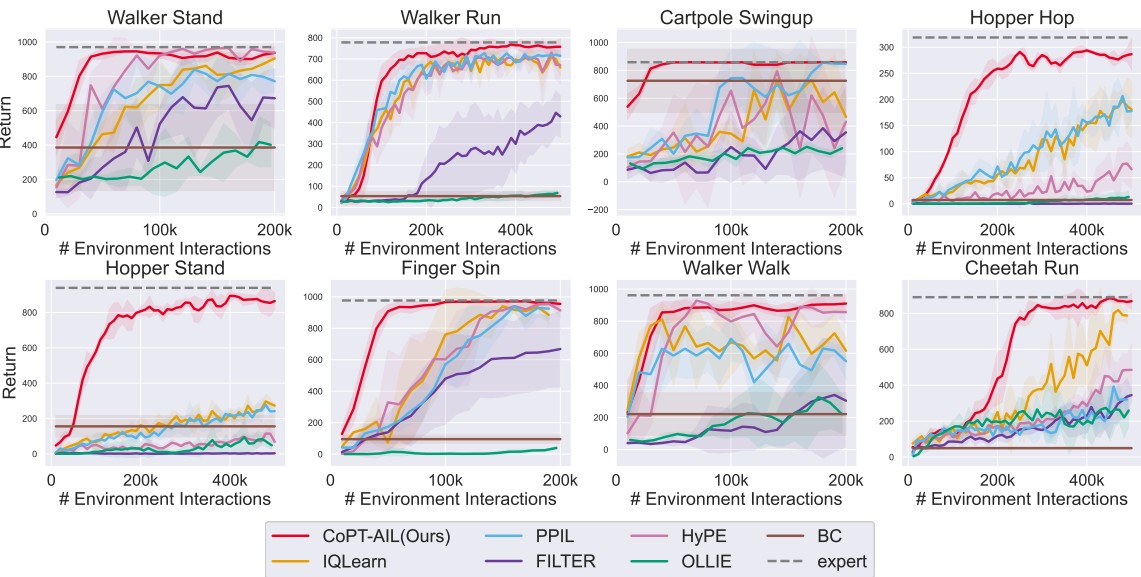

*Figure 2.* Learning curves with respect to online environment interactions on 8 DMControl tasks. Here the $x$-axis is the number of environment interactions and the $y$-axis is the return.

policy and the reward produces a strong initialization that accelerates the subsequent AIL process. To our knowledge, Theorem 1 provides the first theoretical guarantee for the performance benefits of pretraining in AIL.

## 5. Related Work

**Adversarial Imitation Learning.** AIL (Abbeel & Ng, 2004; Syed & Schapire, 2007; Ho & Ermon, 2016; Ghasemipour et al., 2019; Kostrikov et al., 2019) is a prominent class of IL methods that frames imitation as a game-theoretic optimization problem. Despite its high imitation quality, AIL typically requires extensive online environment interaction. To improve interaction efficiency, recent work has explored combining AIL with BC (Jena et al., 2021; Haldar et al., 2023; Watson et al., 2023; Yue et al., 2024). Some approaches augment the AIL objective directly with a BC term (Jena et al., 2021; Haldar et al., 2023), while others incorporate prior policies (Watson et al., 2023) or supplementary datasets (Yue et al., 2024) to warm-start the reward function. However, these methods generally lack theoretical guarantees for the benefits they claim. This paper addresses that gap by providing formal guarantees for the efficiency gains of our proposed approach.

On the theoretical side, several works have analyzed the convergence of AIL in the online setting (Syed & Schapire, 2007; Xu et al., 2021; Shani et al., 2021; Liu et al., 2021; Xu et al., 2023; Viano et al., 2022; 2024; Zhang et al., 2025; Xu et al., 2026a;b). In particular, Shani et al. (2021) propose applying online optimization (Shalev-Shwartz, 2007) to jointly update the policy and reward, and establish an imitation gap

bound in the tabular setting. This line of work has since been extended to function approximation (Liu et al., 2021; Viano et al., 2024; Xu et al., 2026b), and Xu et al. (2026a) recently proved a low-data-regime bound that theoretically explains the strong performance of AIL with limited demonstrations. Notably, all of these works analyze AIL initialized with random policies and rewards, focusing exclusively on the iterative online learning process. None characterizes the error attributable to initialization, nor provides theoretical guarantees for any pretraining strategy. This work fills that gap, offering the first systematic theoretical treatment of reward pretraining in AIL.

**Inverse Reinforcement Learning.** IRL (Ng & Russell, 2000; Arora & Doshi, 2021) aims to recover the underlying reward function from expert demonstrations. Our reward pretraining method is situated within the offline IRL literature (Garg et al., 2021; Yue et al., 2023; Zeng et al., 2023; Wei et al., 2023). Unlike most prior approaches, which require a supplementary non-expert dataset to learn the reward (Yue et al., 2023; Zeng et al., 2023; Wei et al., 2023), our method operates on expert demonstrations alone. While purely offline IRL methods exist (Kostrikov et al., 2020; Garg et al., 2021; Li et al., 2025), our approach is distinctive in that it jointly extracts both the reward function and the policy from a single BC procedure, eliminating the need for a separate IRL step.

## 6. Experiment

This section validates the superiority of CoPT-AIL through experiments. We provide a brief overview of the experi-

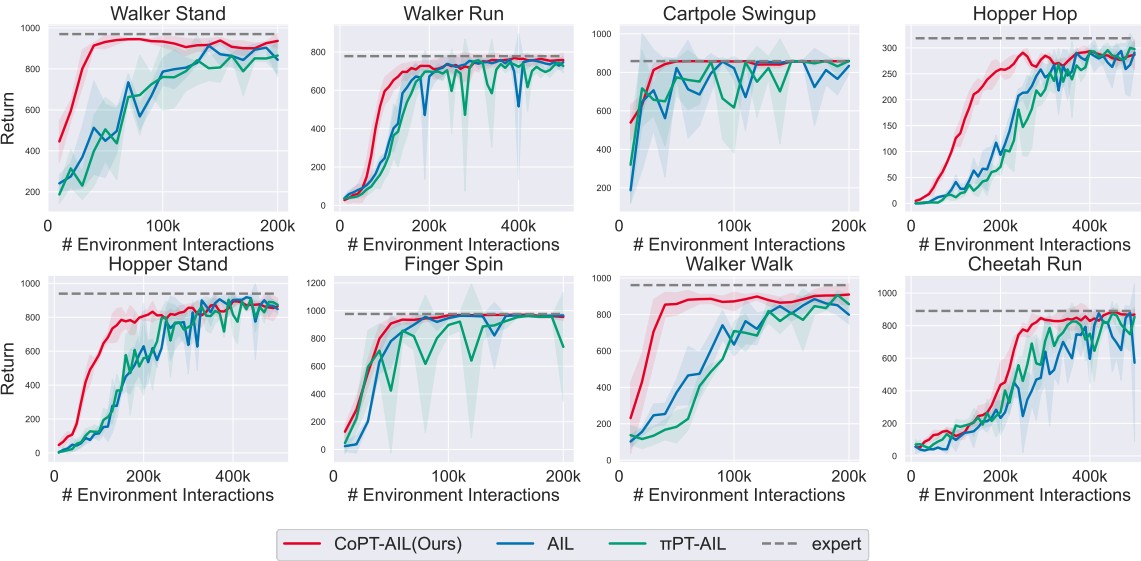

*Figure 3.* Learning curves with respect to online environment interactions on 8 DMControl tasks. Here the $x$-axis is the number of environment interactions and the $y$-axis is the return.

mental setup below, with detailed information available in Appendix C. The implementation of CoPT-AIL is available at https://github.com/LAMDA-RL/CoPT-AIL.

### 6.1. Experiment Setup

**Environment.** We conduct experiments across 8 tasks from the feature-based DMControl benchmark (Tassa et al., 2018), a widely adopted benchmark in imitation learning that provides diverse continuous control tasks. For each task, we train an agent using the online RL algorithm DrQ-v2 (Yarats et al., 2021) with sufficient environment interactions and treat the resulting policy as the expert policy. We then collect expert demonstrations by rolling out this expert policy. Each algorithm is evaluated across three trials with different random seeds, and policy performance is assessed using Monte Carlo approximation over 10 trajectories per evaluation.

**Baselines.** We compare CoPT-AIL against established deep imitation learning methods, including BC (Pomerleau, 1991), IQLearn (Garg et al., 2021), PPIL (Viano et al., 2022), FILTER (Swamy et al., 2023), HyPE (Ren et al., 2024), and OLLIE (Yue et al., 2024)[3], although most lack theoretical guarantees. Notably, FILTER, PPIL, and HyPE represent prior state-of-the-art (SOTA) deep AIL approaches and OL-LIE represents a recent AIL method incorporating reward pretraining. Implementation details are provided in Ap-

pendix C.

### 6.2. Experiment Results

**Overall Performance.** Figure 2 presents the learning curves with respect to online environment interactions for different algorithms. The results reveal that CoPT-AIL consistently matches or exceeds the convergence rates of prior SOTA AIL methods across all 8 tasks. Particularly, on `Cartpole Swingup`, `Hopper Hop`, `Hopper Stand` and `Finger Spin`, CoPT-AIL can achieve near-expert performance with significantly fewer online interactions than existing approaches. These empirical results corroborate our theoretical analysis that the proposed joint pretraining mechanism yields a superior imitation gap in CoPT-AIL.

**Ablation Study.** To validate the effectiveness of our proposed joint pretraining mechanism, we conduct an ablation study comparing CoPT-AIL against two baselines: pure AIL without pretraining (AIL) and AIL with policy pretraining alone ($\pi$PT-AIL). Figure 3 presents the learning curves for these three algorithms. The results reveal that $\pi$PT-AIL achieves convergence rates similar to standard AIL, indicating limited improvement in interaction efficiency from policy pretraining alone. Furthermore, $\pi$PT-AIL exhibits instability, particularly on `Cartpole Swingup` and `Finger Spin` tasks. In contrast, CoPT-AIL demonstrates faster and more stable convergence across 8 tasks, with particularly pronounced improvements on `Walker Stand` and `Walker Walk`.

---

[3]OLLIE originally operates in IL with a supplementary dataset. To ensure a fair comparison within our pure online IL setting, we follow the practice recommended in the OLLIE paper to set the supplementary dataset as an empty set.

# 7. Conclusion

This paper proposes a principled policy-reward joint pretraining method that provably accelerates AIL. We begin with a theoretical analysis of AIL with policy pretraining alone, which isolates reward error as the key theoretical bottleneck and establishes a tight baseline. Guided by this diagnosis, we derive a reward pretraining method grounded in reward-shaping theory, which uncovers a fundamental connection between the expert policy and the shaping reward. This connection naturally gives rise to CoPT-AIL, which jointly pretrains both the policy and the reward through a single BC procedure. Theoretically, CoPT-AIL achieves an improved imitation gap bound over standard AIL without pretraining—providing the first formal guarantee for the benefits of pretraining in AIL. Empirically, CoPT-AIL consistently outperforms prior AIL methods across a range of continuous control tasks.

Looking ahead, several promising directions remain open. First, as an initial step toward understanding the theoretical role of pretraining in AIL, this work focuses on the tabular setting; extending the theoretical results to function approximation is a natural and important next step. Second, it would be interesting to apply CoPT-AIL to more complex robot learning tasks (Tang et al., 2025), particularly those leveraging foundation models such as vision-language-action architectures.

# Acknowledgements

The authors thank Kaiyuan Li for running a baseline. This work was supported by the Yangtze River Delta Science and Technology Innovation Community Joint Research Program (No. 2024CSJZN00302), the Jiangsu Science Foundation (No. BK20243039), the Fundamental and Interdisciplinary Disciplines Breakthrough Plan of the Ministry of Education of China (No. JYB2025XDXM118), and the "111 Center" (No. B26023).

# Impact Statement

This work improves the theoretical understanding and data efficiency of adversarial imitation learning by introducing principled policy–reward co-pretraining. By reducing reliance on extensive environment interaction, our method may benefit real-world applications where data collection is costly or risky. As with other imitation learning approaches, the proposed method may inherit biases from expert demonstrations, highlighting the importance of responsible data collection and evaluation. Overall, we hope this work contributes to more reliable and efficient imitation learning systems.

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

# A. Omitted Proof

## A.1. Proof of Proposition 1

*Proof.* First, we can decompose the imitation gap into the following two terms.

$$V^{\pi^{\mathrm{E}}} - V^{\bar{\pi}} = \frac{1}{K} \sum_{k=1}^{K} \left( V_{r^\star}^{\pi^{\mathrm{E}}} - V_{r^\star}^{\pi^k} \right)$$

$$= \frac{1}{K} \sum_{k=1}^{K} \left( V_{r^\star}^{\pi^{\mathrm{E}}} - V_{r^\star}^{\pi^k} - \left( V_{r^k}^{\pi^{\mathrm{E}}} - V_{r^k}^{\pi^k} \right) \right) + \frac{1}{K} \sum_{k=1}^{K} \left( V_{r^k}^{\pi^{\mathrm{E}}} - V_{r^k}^{\pi^k} \right). \tag{9}$$

We first analyze the first term in the RHS.

$$\frac{1}{K} \sum_{k=1}^{K} \left( V_{r^\star}^{\pi^{\mathrm{E}}} - V_{r^\star}^{\pi^k} - \left( V_{r^k}^{\pi^{\mathrm{E}}} - V_{r^k}^{\pi^k} \right) \right)$$

$$= \frac{1}{K} \left( V_{r^\star}^{\pi^{\mathrm{E}}} - V_{r^\star}^{\pi^1} - \left( V_{r^1}^{\pi^{\mathrm{E}}} - V_{r^1}^{\pi^1} \right) \right) + \frac{1}{K} \sum_{k=2}^{K} \left( \widehat{V}_{r^\star}^{\pi^{\mathrm{E}}} - V_{r^\star}^{\pi^k} - \left( \widehat{V}_{r^k}^{\pi^{\mathrm{E}}} - V_{r^k}^{\pi^k} \right) \right) + V_{r^\star}^{\pi^{\mathrm{E}}} - \widehat{V}_{r^\star}^{\pi^{\mathrm{E}}}$$

$$+ \frac{1}{K} \sum_{k=2}^{K} \left( \widehat{V}_{r^k}^{\pi^{\mathrm{E}}} - V_{r^k}^{\pi^{\mathrm{E}}} \right)$$

$$\leq \frac{1}{K} \left( V_{r^\star}^{\pi^{\mathrm{E}}} - V_{r^\star}^{\pi^1} - \left( V_{r^1}^{\pi^{\mathrm{E}}} - V_{r^1}^{\pi^1} \right) \right) + \frac{1}{K} \sum_{k=2}^{K} \left( \widehat{V}_{r^\star}^{\pi^{\mathrm{E}}} - V_{r^\star}^{\pi^k} - \left( \widehat{V}_{r^k}^{\pi^{\mathrm{E}}} - V_{r^k}^{\pi^k} \right) \right)$$

$$+ 2 \max_{r \in \mathcal{R}} \left| V_r^{\pi^{\mathrm{E}}} - \widehat{V}_r^{\pi^{\mathrm{E}}} \right|.$$

For any reward $r \in \mathcal{R}$, $\widehat{V}_r^{\pi^{\mathrm{E}}} := \mathbb{E}_{\tau \sim \mathcal{D}^{\mathrm{E}}} \left[ \sum_{h=1}^{H} r_h(s_h, a_h) \right]$ denotes the empirical estimation of $V_r^{\pi^{\mathrm{E}}}$ based on demonstrations $\mathcal{D}^{\mathrm{E}}$. Furthermore, $\forall r \in \mathcal{R}$, we have that

$$\left| V_r^{\pi^{\mathrm{E}}} - \widehat{V}_r^{\pi^{\mathrm{E}}} \right| = \left| \sum_{h=1}^{H} \sum_{(s,a) \in \mathcal{S} \times \mathcal{A}} \left( d_h^{\pi^{\mathrm{E}}}(s,a) - \widehat{d_h^{\pi^{\mathrm{E}}}}(s,a) \right) r_h(s,a) \right|$$

$$\leq \sum_{h=1}^{H} \sum_{(s,a) \in \mathcal{S} \times \mathcal{A}} \left| d_h^{\pi^{\mathrm{E}}}(s,a) - \widehat{d_h^{\pi^{\mathrm{E}}}}(s,a) \right| \left| r_h(s,a) \right|$$

$$\leq \sum_{h=1}^{H} \sum_{(s,a) \in \mathcal{S} \times \mathcal{A}} \left| d_h^{\pi^{\mathrm{E}}}(s,a) - \widehat{d_h^{\pi^{\mathrm{E}}}}(s,a) \right|.$$

Here $d_h^{\pi^{\mathrm{E}}}(s,a) := \mathbb{P}^{\pi^{\mathrm{E}}}(s_h = s, a_h = a)$ represents the probability of visiting $(s,a)$ in time step $h$ by following $\pi^{\mathrm{E}}$. Besides, $\widehat{d_h^{\pi^{\mathrm{E}}}}(s,a) := n_h^{\mathrm{E}}(s,a)/N$ represents the empirical estimation based on demonstrations $\mathcal{D}^{\mathrm{E}}$, where $n_h^{\mathrm{E}}(s,a)$ denotes the number of times that $(s,a)$ is visited in time step $h$ in $\mathcal{D}^{\mathrm{E}}$. The last inequality follows that $r_h(s,a) \in [0,1]$.

Based on (Weissman et al., 2003) and union bound, for any $\delta \in (0,1)$, with probability at least $1 - \delta$, we have that

$$\forall h \in [H], \quad \sum_{(s,a) \in \mathcal{S} \times \mathcal{A}} \left| d_h^{\pi^{\mathrm{E}}}(s,a) - \widehat{d_h^{\pi^{\mathrm{E}}}}(s,a) \right| \leq \sqrt{\frac{2|\mathcal{S}||\mathcal{A}| \log(H/\delta)}{N}}.$$

Then it holds that

$$\forall r \in \mathcal{R}, \ \left| V_r^{\pi^{\mathrm{E}}} - \widehat{V}_r^{\pi^{\mathrm{E}}} \right| \leq H \sqrt{\frac{2|\mathcal{S}||\mathcal{A}| \log(H/\delta)}{N}}.$$

Then we can obtain the following upper bound.

$$\frac{1}{K}\sum_{k=1}^{K}\left(V_{r^\star}^{\pi^{\mathrm{E}}} - V_{r^\star}^{\pi^k} - \left(V_{r^k}^{\pi^{\mathrm{E}}} - V_{r^k}^{\pi^k}\right)\right)$$

$$\leq \frac{1}{K}\left(V_{r^\star}^{\pi^{\mathrm{E}}} - V_{r^\star}^{\pi^1} - \left(V_{r^1}^{\pi^{\mathrm{E}}} - V_{r^1}^{\pi^1}\right)\right) + \frac{1}{K}\sum_{k=2}^{K}\left(\widehat{V}_{r^\star}^{\pi^{\mathrm{E}}} - V_{r^\star}^{\pi^k} - \left(\widehat{V}_{r^k}^{\pi^{\mathrm{E}}} - V_{r^k}^{\pi^k}\right)\right)$$

$$+ 2H\sqrt{\frac{2|\mathcal{S}||\mathcal{A}|\log(H/\delta)}{N}}$$

$$\leq \frac{1}{K}\left(V_{r^\star}^{\pi^{\mathrm{E}}} - V_{r^\star}^{\pi^1} - \left(V_{r^1}^{\pi^{\mathrm{E}}} - V_{r^1}^{\pi^1}\right)\right) + 2H\sqrt{\frac{2|\mathcal{S}||\mathcal{A}|\log(H/\delta)}{N}}. \tag{10}$$

The last inequality follows that $\forall k \geq 2, r^k = \operatorname{argmin}_{r\in\mathcal{R}} V_r^{\pi^k} - \widehat{V}_r^{\pi^{\mathrm{E}}}$.

We proceed to analyze the second term in the RHS of Eq.(9). According to the policy difference lemma (Kakade & Langford, 2002), we can obtain that

$$\frac{1}{K}\sum_{k=1}^{K}\left(V_{r^k}^{\pi^{\mathrm{E}}} - V_{r^k}^{\pi^k}\right) = \frac{1}{K}\sum_{k=1}^{K}\mathbb{E}\left[\sum_{h=1}^{H}\langle Q_h^{\pi^k,r^k}(s_h,\cdot), \pi_h^{\mathrm{E}}(\cdot|s_h) - \pi_h^k(\cdot|s_h)\rangle\Big|\pi^{\mathrm{E}}\right]$$

$$= \frac{1}{K}\mathbb{E}\left[\sum_{h=1}^{H}\sum_{k=1}^{K}\langle Q_h^{\pi^k,r^k}(s_h,\cdot), \pi_h^{\mathrm{E}}(\cdot|s_h) - \pi_h^k(\cdot|s_h)\rangle\Big|\pi^{\mathrm{E}}\right].$$

For each $(s,h)\in\mathcal{S}\times[H]$, we analyze the error term of $\sum_{k=1}^{K}\langle Q_h^{\pi^k,r^k}(s,\cdot), \pi_h^{\mathrm{E}}(\cdot|s) - \pi_h^k(\cdot|s)\rangle$. For a simplex $p\in\Delta(\mathcal{A})$, we define the linear function $\ell_{s,h}^k(p) := -\sum_{a\in\mathcal{A}} p(a)Q_h^{\pi^k,r^k}(s,a)$. Then we can regard the above error term as the regret for the online optimization problem with loss functions $\{\ell_{s,h}^k(p)\}_{k=1}^{K}$.

$$\sum_{k=1}^{K}\langle Q_h^{\pi^k,r^k}(s,\cdot), \pi_h^{\mathrm{E}}(\cdot|s) - \pi_h^k(\cdot|s)\rangle = \sum_{k=1}^{K}\ell_{s,h}^k(\pi_h^k(\cdot|s)) - \ell_{s,h}^k(\pi_h^{\mathrm{E}}(\cdot|s)).$$

Furthermore, performing KL-regularized policy optimization is equivalent to applying online mirror descent (Orabona, 2019) on the loss functions $\{\ell_{s,h}^k(p)\}_{k=1}^{K}$. According to the regret bound on online mirror descent (e.g., (Orabona, 2019, Theorem 6.8)), we have that

$$\sum_{k=1}^{K}\ell_{s,h}^k(\pi_h^k(\cdot|s)) - \ell_{s,h}^k(\pi_h^{\mathrm{E}}(\cdot|s)) \leq \frac{D_{\mathrm{KL}}(\pi_h^{\mathrm{E}}(\cdot|s), \pi_h^1(\cdot|s))}{\eta} + \frac{\eta}{2}\sum_{k=1}^{K}\left\|Q_h^{\pi^k,r^k}(s,\cdot)\right\|_\infty^2$$

$$\leq \frac{D_{\mathrm{KL}}(\pi_h^{\mathrm{E}}(\cdot|s), \pi_h^1(\cdot|s))}{\eta} + \frac{\eta}{2}KH^2.$$

The last inequality follows that $Q_h^{\pi^k,r^k}(s,a)\in[0,H]$ because of $r_h^k(s,a)\in[0,1], \forall(s,a,h)\in\mathcal{S}\times\mathcal{A}\times[H]$. Then we can obtain that

$$\frac{1}{K}\sum_{k=1}^{K}\left(V_{r^k}^{\pi^{\mathrm{E}}} - V_{r^k}^{\pi^k}\right) = \frac{1}{K}\mathbb{E}\left[\sum_{h=1}^{H}\sum_{k=1}^{K}\langle Q_h^{\pi^k,r^k}(s_h,\cdot), \pi_h^{\mathrm{E}}(\cdot|s_h) - \pi^k(\cdot|s_h)\rangle\Big|\pi^{\mathrm{E}}\right]$$

$$\leq \frac{1}{K}\mathbb{E}\left[\sum_{h=1}^{H}\frac{D_{\mathrm{KL}}(\pi_h^{\mathrm{E}}(\cdot|s_h), \pi_h^1(\cdot|s_h))}{\eta} + \frac{\eta}{2}KH^2\Big|\pi^{\mathrm{E}}\right]$$

$$= \frac{1}{\eta K}\mathbb{E}\left[\sum_{h=1}^{H} D_{\mathrm{KL}}(\pi_h^{\mathrm{E}}(\cdot|s_h), \pi_h^1(\cdot|s_h))\Big|\pi^{\mathrm{E}}\right] + \frac{\eta}{2}H^3$$

$$= \frac{1}{\eta K}\mathbb{E}\left[\sum_{h=1}^{H} D_{\mathrm{KL}}(\pi_h^{\mathrm{E}}(\cdot|s_h), \pi_h^{\mathrm{BC}}(\cdot|s_h))\Big|\pi^{\mathrm{E}}\right] + \frac{\eta}{2}H^3. \tag{11}$$

Combining the bounds in Eq.(10) and Eq.(11) finishes the proof of Eq.(4).

$$V^{\pi^{\mathrm{E}}} - V^{\bar{\pi}} \le \frac{1}{K} \left( V^{\pi^{\mathrm{E}}}_{r^\star} - V^{\pi^1}_{r^\star} - \left( V^{\pi^{\mathrm{E}}}_{r^1} - V^{\pi^1}_{r^1} \right) \right) + 2H\sqrt{\frac{2|\mathcal{S}||\mathcal{A}|\log(H/\delta)}{N}}$$
$$+ \frac{1}{\eta K}\mathbb{E}\left[ \sum_{h=1}^{H} D_{\mathrm{KL}}(\pi^{\mathrm{E}}_h(\cdot|s_h), \pi^{\mathrm{BC}}_h(\cdot|s_h)) \Big| \pi^{\mathrm{E}} \right] + \frac{\eta}{2}H^3.$$

We proceed to prove Eq.(5) in Proposition 1. In particular, with the policy pre-trained via BC, we can leverage the guarantee of BC to analyze the KL divergence between the expert policy and the initial policy. Note that we have proved the following upper bound.

$$\frac{1}{K}\sum_{k=1}^{K} \left( V^{\pi^{\mathrm{E}}}_{r^k} - V^{\pi^k}_{r^k} \right) \le \frac{1}{\eta K}\mathbb{E}\left[ \sum_{h=1}^{H} D_{\mathrm{KL}}(\pi^{\mathrm{E}}_h(\cdot|s_h), \pi^{\mathrm{BC}}_h(\cdot|s_h)) \Big| \pi^{\mathrm{E}} \right] + \frac{\eta}{2}H^3.$$

According to (Tiapkin et al., 2024, Corollary 1), with probability at least $1 - \delta$, we have that

$$\frac{1}{K}\sum_{k=1}^{K} \left( V^{\pi^{\mathrm{E}}}_{r^k} - V^{\pi^k}_{r^k} \right) \le \frac{6|\mathcal{S}||\mathcal{A}|H \cdot \log\left(2\mathrm{e}^4 N\right) \cdot \log\left(12HN^2/\delta\right)}{\eta KN} + \frac{18AH}{\eta KN} + \frac{\eta}{2}H^3$$
$$\le \frac{24|\mathcal{S}||\mathcal{A}|H \cdot \log\left(2\mathrm{e}^4 N\right) \cdot \log\left(12HN^2/\delta\right)}{\eta KN} + \frac{\eta}{2}H^3$$
$$\le \frac{24|\mathcal{S}||\mathcal{A}|H \log^2(2e^4 HN^2/\delta)}{\eta KN} + \frac{\eta}{2}H^3$$
$$= 4\sqrt{\frac{3|\mathcal{S}||\mathcal{A}|H^4 \log^2(2e^4 HN^2/\delta)}{KN}}.$$

The last equation holds by choosing the step-size $\eta = \sqrt{(48|\mathcal{S}||\mathcal{A}|\log^2(2e^4 HN^2/\delta))/(H^2 KN)}$. Finally, by union bound, we have that

$$V^{\pi^{\mathrm{E}}} - V^{\bar{\pi}} \le \frac{1}{K} \left( V^{\pi^{\mathrm{E}}}_{r^\star} - V^{\pi^1}_{r^\star} - \left( V^{\pi^{\mathrm{E}}}_{r^1} - V^{\pi^1}_{r^1} \right) \right) + 2H\sqrt{\frac{2|\mathcal{S}||\mathcal{A}|\log(2H/\delta)}{N}}$$
$$+ 4\sqrt{\frac{3|\mathcal{S}||\mathcal{A}|H^4 \log^2(4e^4 HN^2/\delta)}{KN}}.$$

We complete the proof.

$\square$

## A.2. Proof of Proposition 2

Recall the definition of shaping reward $\widetilde{r}_h(s,a) := r_h(s,a) - \Phi_h(s) + \mathbb{E}_{s'\sim P_h(\cdot|s,a)}[\Phi_{h+1}(s')]$. For any policy $\pi$, we have that

$$V^{\pi}_{\widetilde{r}} = \mathbb{E}\left[ \sum_{h=1}^{H} \widetilde{r}_h(s_h, a_h) \Big| \pi \right]$$
$$= \mathbb{E}\left[ \sum_{h=1}^{H} \left( r_h(s_h, a_h) - \Phi_h(s_h) + \mathbb{E}_{s'\sim P_h(\cdot|s_h,a_h)}[\Phi_{h+1}(s')] \right) \Big| \pi \right]$$
$$\stackrel{(a)}{=} \mathbb{E}\left[ \sum_{h=1}^{H} \left( r_h(s_h, a_h) - \Phi_h(s_h) + \Phi_{h+1}(s_{h+1}) \right) \Big| \pi \right]$$
$$\stackrel{(b)}{=} \mathbb{E}\left[ \sum_{h=1}^{H} r_h(s_h, a_h) - \Phi_1(s_1) \Big| \pi \right]$$

$$= \mathbb{E}\left[\sum_{h=1}^{H} r_h(s_h, a_h)\Big|\pi\right] - \mathbb{E}_{s_1 \sim \rho}\left[\Phi(s_1)\right]$$

$$= V_r^\pi - \mathbb{E}_{s_1 \sim \rho}\left[\Phi(s_1)\right].$$

Here Equation (a) follows the tower property and $s_{h+1} \sim P_h(\cdot|s_h, a_h)$. Equation (b) follows the telescoping argument with boundary condition $\Phi_{H+1} \equiv 0$. Then for any pair of policies $\pi$ and $\pi'$, it holds that

$$V_r^{\pi'} - V_r^\pi = (V_{\widetilde{r}}^{\pi'} + \mathbb{E}_{s_1 \sim \rho}[\Phi(s_1)]) - (V_{\widetilde{r}}^\pi + \mathbb{E}_{s_1 \sim \rho}[\Phi(s_1)]) = V_{\widetilde{r}}^{\pi'} - V_{\widetilde{r}}^\pi.$$

We complete the proof.

### A.3. Proof of Theorem 1

We first analyze the reward error of $(1/K) \cdot (V_{r^\star}^{\pi^E} - V_{r^\star}^{\pi^1} - (V_{r^1}^{\pi^E} - V_{r^1}^{\pi^1}))$. Recall that $\widetilde{r}_h^\star(s, a) := \log(\pi_h^E(a|s))$ is exactly a shaping reward of $r_h^\star(s, a)$ regarding the potential-based shaping functions $\{V_h^\star\}$. According to 2, we can obtain that

$$V_{r^\star}^{\pi^E} - V_{r^\star}^{\pi^1} = V_{\widetilde{r}^\star}^{\pi^E} - V_{\widetilde{r}^\star}^{\pi^1}$$

$$= \mathbb{E}\left[\sum_{h=1}^{H} \log(\pi_h^E(a_h|s_h))\Big|\pi^E\right] - \mathbb{E}\left[\sum_{h=1}^{H} \log(\pi_h^E(a_h|s_h))\Big|\pi^1\right].$$

Then we can obtain that

$$\left(V_{r^\star}^{\pi^E} - V_{r^\star}^{\pi^1} - \left(V_{r^1}^{\pi^E} - V_{r^1}^{\pi^1}\right)\right)$$

$$= \left(\mathbb{E}\left[\sum_{h=1}^{H} \log(\pi_h^E(a_h|s_h))\Big|\pi^E\right] - \mathbb{E}\left[\sum_{h=1}^{H} \log(\pi_h^E(a_h|s_h))\Big|\pi^1\right]\right.$$

$$\left. - \left(\mathbb{E}\left[\sum_{h=1}^{H} \log(\pi_h^{BC}(a_h|s_h))\Big|\pi^E\right] - \mathbb{E}\left[\sum_{h=1}^{H} \log(\pi_h^{BC}(a_h|s_h))\Big|\pi^1\right]\right)\right)$$

$$= \mathbb{E}\left[\sum_{h=1}^{H} \log(\pi_h^E(a_h|s_h)) - \log(\pi_h^{BC}(a_h|s_h))\Big|\pi^E\right]$$

$$- \mathbb{E}\left[\sum_{h=1}^{H} \log(\pi_h^E(a_h|s_h)) - \log(\pi_h^{BC}(a_h|s_h))\Big|\pi^1\right]$$

$$= \mathbb{E}\left[\sum_{h=1}^{H} D_{KL}(\pi_h^E(\cdot|s_h), \pi_h^{BC}(\cdot|s_h))\Big|\pi^E\right] + \mathbb{E}\left[\sum_{h=1}^{H} D_{KL}(\pi_h^{BC}(\cdot|s_h), \pi_h^E(\cdot|s_h))\Big|\pi^{BC}\right].$$

Based on (Tiapkin et al., 2024, Corollary 1), we can upper bound the first term in the RHS. With probability at least $1 - \delta$, we have that

$$\mathbb{E}\left[\sum_{h=1}^{H} D_{KL}(\pi_h^E(\cdot|s_h), \pi_h^{BC}(\cdot|s_h))\Big|\pi^E\right]$$

$$\leq \frac{6|\mathcal{S}||\mathcal{A}|H \cdot \log\left(2e^4 N\right) \cdot \log\left(12HN^2/\delta\right)}{N} + \frac{18|\mathcal{A}|H}{N}$$

$$\leq \frac{24|\mathcal{S}||\mathcal{A}|H \cdot \log^2\left(2e^4 HN^2/\delta\right)}{N}.$$

We further upper bound the second term.

$$\mathbb{E}\left[\sum_{h=1}^{H} D_{KL}(\pi_h^{BC}(\cdot|s_h), \pi_h^E(\cdot|s_h))\Big|\pi^{BC}\right] = \sum_{h=1}^{H} \sum_{s \in \mathcal{S}} d_h^{\pi^{BC}}(s) D_{KL}(\pi_h^{BC}(\cdot|s), \pi_h^E(\cdot|s))$$

$$\leq C \sum_{h=1}^{H} \sum_{s \in \mathcal{S}} d_h^{\pi^{\mathrm{E}}}(s) D_{\mathrm{KL}}(\pi_h^{\mathrm{BC}}(\cdot|s), \pi_h^{\mathrm{E}}(\cdot|s))$$

Here $C := \max_{(s,h) \in \mathcal{S} \times [H]} d_h^{\pi^{\mathrm{BC}}}(s)/d_h^{\pi^{\mathrm{E}}}(s)$. According to Lemma 1, with probability at least $1 - \delta$, it holds that

$$\forall (s,h) \in \mathcal{S} \times [H], \ D_{\mathrm{KL}}(\pi_h^{\mathrm{BC}}(\cdot|s), \pi_h^{\mathrm{E}}(\cdot|s)) \leq \frac{H|\mathcal{A}|\log(4|\mathcal{S}||\mathcal{A}|H(N+1)/\delta)}{N_h(s) + |\mathcal{A}|}.$$

Besides, with Lemma 4 and union bound, with probability at least $1 - \delta$,

$$\forall (s,h) \in \mathcal{S} \times [H], \frac{d_h^{\pi^{\mathrm{E}}}(s)}{\max\{N_h(s), 1\}} \leq \frac{12 \log(2|\mathcal{S}|H/\delta)}{N}.$$

With union bound, the above two events happen with probability at least $1 - 2\delta$. Conditioned on these two events, we have that

$$\mathbb{E}\left[\sum_{h=1}^{H} D_{\mathrm{KL}}(\pi_h^{\mathrm{BC}}(\cdot|s_h), \pi_h^{\mathrm{E}}(\cdot|s_h)) \middle| \pi^{\mathrm{BC}}\right]$$

$$\leq C \sum_{h=1}^{H} \sum_{s \in \mathcal{S}} d_h^{\pi^{\mathrm{E}}}(s) \frac{|\mathcal{A}|H \log(4|\mathcal{S}||\mathcal{A}|H(N+1)/\delta)}{N_h(s) + |\mathcal{A}|}$$

$$= C|\mathcal{A}|H \log(4|\mathcal{S}||\mathcal{A}|H(N+1)/\delta) \sum_{h=1}^{H} \sum_{s \in \mathcal{S}} \frac{d_h^{\pi^{\mathrm{E}}}(s)}{N_h(s) + |\mathcal{A}|}$$

$$\leq C|\mathcal{A}|H \log(4|\mathcal{S}||\mathcal{A}|H(N+1)/\delta) \sum_{h=1}^{H} \sum_{s \in \mathcal{S}} \frac{d_h^{\pi^{\mathrm{E}}}(s)}{\max\{N_h(s), 1\}}$$

$$\leq 12 \frac{C|\mathcal{S}||\mathcal{A}|H^2 \log(4|\mathcal{S}||\mathcal{A}|H(N+1)/\delta) \log(2|\mathcal{S}|H/\delta)}{N}$$

$$\leq 12 \frac{C|\mathcal{S}||\mathcal{A}|H^2 \log^2(4|\mathcal{S}||\mathcal{A}|H(N+1)/\delta)}{N}.$$

By union bound, with probability at least $1 - \delta$, it holds that

$$\frac{1}{K}\left(V_{r^\star}^{\pi^{\mathrm{E}}} - V_{r^\star}^{\pi^1} - \left(V_{r^1}^{\pi^{\mathrm{E}}} - V_{r^1}^{\pi^1}\right)\right)$$

$$\leq \frac{24|\mathcal{S}||\mathcal{A}|H \cdot \log^2\left(6e^4 HN^2/\delta\right)}{KN} + 12 \frac{C|\mathcal{S}||\mathcal{A}|H^2 \log^2(12|\mathcal{S}||\mathcal{A}|H(N+1)/\delta)}{KN}$$

$$\leq 48 \frac{C|\mathcal{S}||\mathcal{A}|H^2 \log^2\left(6e^4|\mathcal{S}||\mathcal{A}|HN^2/\delta\right)}{KN}.$$

We complete the proof of Eq.(7). Furthermore, Algorithm 2 differs from Algorithm 1 only in the reward initialization. Therefore, by following the same analysis in the proof of Proposition 1, we can obtain that

$$V^{\pi^{\mathrm{E}}} - V^{\bar{\pi}} \leq \frac{1}{K}\left(V_{r^\star}^{\pi^{\mathrm{E}}} - V_{r^\star}^{\pi^1} - \left(V_{r^1}^{\pi^{\mathrm{E}}} - V_{r^1}^{\pi^1}\right)\right) + 4\sqrt{\frac{3|\mathcal{S}||\mathcal{A}|H^4 \log^2(4e^4 HN^2/\delta)}{KN}}$$

$$+ 2H\sqrt{\frac{2|\mathcal{S}||\mathcal{A}| \log(2H/\delta)}{N}}$$

$$\leq 48 \frac{C|\mathcal{S}||\mathcal{A}|H^2 \log^2\left(6e^4|\mathcal{S}||\mathcal{A}|HN^2/\delta\right)}{KN} + 4\sqrt{\frac{3|\mathcal{S}||\mathcal{A}|H^4 \log^2(4e^4 HN^2/\delta)}{KN}}$$

$$+ 2H\sqrt{\frac{2|\mathcal{S}||\mathcal{A}| \log(2H/\delta)}{N}}.$$

We complete the proof of Eq.(8).

## A.4. Bound Comparison

In this part, we compare the imitation gap bound of CoPT-AILwith that of OAL (Shani et al., 2021), a standard AIL algorithm without pretraining. In particular, without any pretraining, OAL updates the reward function via online projected gradient descent (Orabona, 2019) and updates the policy via KL-regularized policy optimization. Similar to CoPT-AIL, we consider that OAL can compute the Q-value function of the current policy. Now, we are ready to perform the bound comparison. Shani et al. (2021) decomposes the imitation gap into the following terms.

$$V_{r^\star}^{\pi^E} - V_{r^\star}^{\bar{\pi}} \leq \underbrace{\frac{1}{K} \sum_{k=1}^{K} \left( \left( \widehat{V}_{r^\star}^{\pi^E} - V_{r^\star}^{\pi^k} \right) - \left( \widehat{V}_{r^k}^{\pi^E} - V_{r^k}^{\pi^k} \right) \right)}_{:=\text{Term I}} + \underbrace{\frac{1}{K} \sum_{k=1}^{K} \left( V_{r^k}^{\pi^E} - V_{r^k}^{\pi^k} \right)}_{:=\text{Term II}}$$
$$+ \underbrace{2 \max_{r \in \mathcal{R}} \left| \widehat{V}_r^{\pi^E} - V_r^{\pi^E} \right|}_{:=\text{Term III}}.$$

Lemmas 4, 5, and 6 in (Shani et al., 2021) upper bound Terms I, II, and III, respectively.

$$\text{Term I} \precsim \sqrt{\frac{|\mathcal{S}||\mathcal{A}|H^2}{K}}, \ \text{Term II} \precsim \sqrt{\frac{H^4 \log(|\mathcal{A}|)}{K}}, \ \text{Term III} \precsim \sqrt{\frac{|\mathcal{S}||\mathcal{A}|H^3 \log(1/\delta)}{N}}.$$

Finally, OAL attains the imitation gap bound of

$$\widetilde{\mathcal{O}} \left\{ \min \left\{ \sqrt{\frac{|\mathcal{S}||\mathcal{A}|H^2}{K}} + \sqrt{\frac{H^4 \log(|\mathcal{A}|)}{K}} + \sqrt{\frac{|\mathcal{S}||\mathcal{A}|H^3}{N}}, H \right\} \right\}.$$

Here $H$ represents the maximum value for the imitation gap. In comparison, CoPT-AIL attains the bound of

$$\widetilde{\mathcal{O}} \left( \min \left\{ \frac{C|\mathcal{S}||\mathcal{A}|H^2}{KN} + \sqrt{\frac{|\mathcal{S}||\mathcal{A}|H^4}{KN}} + \sqrt{\frac{|\mathcal{S}||\mathcal{A}|H^2}{N}}, H \right\} \right).$$

It is direct to derive that CoPT-AIL can achieve an improved imitation gap bound when $N \succsim C\sqrt{|\mathcal{S}||\mathcal{A}|H^2/K}$.

# B. Useful Lemmas

First, we provide the basic theoretical guarantee on BC. Following (Tiapkin et al., 2024), we consider the BC algorithm formulated as

$$\pi^{\text{BC}} \in \underset{\pi \in \Pi}{\arg\max} \sum_{h=1}^{H} \left( \sum_{i=1}^{N} \log(\pi_h(a_h^i \mid s_h^i)) + \mathcal{R}_h(\pi_h) \right). \tag{12}$$

Here $\mathcal{D} = \{(s_1^i, a_1^i, \ldots, s_H^i, a_H^i)\}_{i=1}^{N}$ denotes expert demonstrations and $\mathcal{R}_h(\pi_h) = \sum_{(s,a) \in \mathcal{S} \times \mathcal{A}} \log(\pi_h(a|s))$ is the regularizer. Tiapkin et al. (2024) proved theoretical bounds on the forward KL divergence between $\pi^E$ and $\pi^{\text{BC}}$. In the sequel, we provide a bound on the reverse KL divergence, which could be of independent interest.

**Lemma 1.** *Consider Eq.(12). With probability at least $1 - \delta$, it holds that*

$$\forall (s, h) \in \mathcal{S} \times [H], \ D_{\text{KL}}(\pi_h^{\text{BC}}(\cdot|s), \pi_h^E(\cdot|s)) \leq \frac{H|\mathcal{A}| \log(4|\mathcal{S}||\mathcal{A}|H(N+1)/\delta)}{N_h(s) + |\mathcal{A}|}.$$

*Here $N_h(s) := \sum_{i=1}^{N} \mathbb{I}\{s_h^i = s\}$ denotes the number of times that states $s$ appears in demonstrations.*

*Proof.* The optimization problem in Eq.(12) admits the closed-form solution of

$$\pi_h^{\text{BC}}(a|s) = \frac{N_h(s, a) + 1}{N_h(s) + |\mathcal{A}|}. \tag{13}$$

Here $N_h(s, a)$ represents the number of times that the state-action pair $(s, a)$ is visited in $\mathcal{D}$. We first analyze the case where $N_h(s) > 0$. We aim to upper bound the probability of $\mathbb{P}(D_{\mathrm{KL}}(\pi_h^{\mathrm{BC}}(\cdot|s), \pi_h^{\mathrm{E}}(\cdot|s)) \geq \varepsilon)$ for each $(s, h) \in \mathcal{S} \times [H]$. To analyze this probability, we reformulate $\pi^{\mathrm{BC}}$ as a mixture of two distributions.

$$\pi_h^{\mathrm{BC}}(a|s) = \frac{N_h(s)}{N_h(s) + |\mathcal{A}|} \cdot \frac{N_h(s, a)}{N_h(s)} + \frac{|\mathcal{A}|}{N_h(s) + |\mathcal{A}|} \cdot \frac{1}{|\mathcal{A}|}$$

$$= \frac{N_h(s)}{N_h(s) + |\mathcal{A}|} \cdot \widehat{\pi}_h(a|s) + \frac{|\mathcal{A}|}{N_h(s) + |\mathcal{A}|} \cdot p(a).$$

Here $\hat{\pi}$ denotes the empirical distribution from $\mathcal{D}$ and $p$ denotes the uniform distribution over $\mathcal{A}$. Furthermore, based on the convexity of KL divergence, we have that

$$D_{\mathrm{KL}}(\pi_h^{\mathrm{BC}}(\cdot|s), \pi_h^{\mathrm{E}}(\cdot|s)) \leq \frac{N_h(s)}{N_h(s) + |\mathcal{A}|} D_{\mathrm{KL}}(\widehat{\pi}_h(\cdot|s), \pi_h^{\mathrm{E}}(\cdot|s))$$

$$+ \frac{|\mathcal{A}|}{N_h(s) + |\mathcal{A}|} D_{\mathrm{KL}}(p(\cdot), \pi_h^{\mathrm{E}}(\cdot|s)).$$

Therefore, the event of $D_{\mathrm{KL}}(\pi_h^{\mathrm{BC}}(\cdot|s), \pi_h^{\mathrm{E}}(\cdot|s)) \geq \varepsilon$ implies that

$$D_{\mathrm{KL}}(\widehat{\pi}_h(\cdot|s), \pi_h^{\mathrm{E}}(\cdot|s)) \geq \frac{N_h(s) + |\mathcal{A}|}{N_h(s)} \cdot \left( \varepsilon - \frac{|\mathcal{A}|}{N_h(s) + |\mathcal{A}|} D_{\mathrm{KL}}(p(\cdot), \pi_h^{\mathrm{E}}(\cdot|s)) \right).$$

We define that

$$\varepsilon' := \frac{N_h(s) + |\mathcal{A}|}{N_h(s)} \cdot \left( \varepsilon - \frac{|\mathcal{A}|}{N_h(s) + |\mathcal{A}|} D_{\mathrm{KL}}(p(\cdot), \pi_h^{\mathrm{E}}(\cdot|s)) \right).$$

Then we have that

$$\mathbb{P}(D_{\mathrm{KL}}(\pi_h^{\mathrm{BC}}(\cdot|s), \pi_h^{\mathrm{E}}(\cdot|s)) \geq \varepsilon) \leq \mathbb{P}(D_{\mathrm{KL}}(\widehat{\pi}_h(a|s), \pi_h^{\mathrm{E}}(\cdot|s)) \geq \varepsilon').$$

According to Sanov's Theorem (Lemma 2), we have that

$$\mathbb{P}(D_{\mathrm{KL}}(\widehat{\pi}_h(a|s), \pi_h^{\mathrm{E}}(\cdot|s)) \geq \varepsilon') \leq (N_h(s) + 1)^{|\mathcal{A}|} \exp(-N_h(s)\varepsilon').$$

Setting the term in the RHS as the failure probability $\delta$ yields that

$$\varepsilon' = \frac{|\mathcal{A}| \log(N_h(s) + 1) + \log(1/\delta)}{N_h(s)},$$

$$\varepsilon = \frac{|\mathcal{A}| \log(N_h(s) + 1) + \log(1/\delta) + |\mathcal{A}| D_{\mathrm{KL}}(p(\cdot), \pi_h^{\mathrm{E}}(\cdot|s))}{N_h(s) + |\mathcal{A}|}.$$

This implies that for $N_h(s) > 0$, with probability at least $1 - \delta$,

$$D_{\mathrm{KL}}(\pi_h^{\mathrm{BC}}(\cdot|s), \pi_h^{\mathrm{E}}(\cdot|s)) \leq \frac{|\mathcal{A}| \log(N_h(s) + 1) + \log(1/\delta) + |\mathcal{A}| D_{\mathrm{KL}}(p(\cdot), \pi_h^{\mathrm{E}}(\cdot|s))}{N_h(s) + |\mathcal{A}|}$$

$$\overset{\text{(a)}}{\leq} \frac{|\mathcal{A}| \log(N_h(s) + 1) + \log(1/\delta) + H|\mathcal{A}| \log(4|\mathcal{A}|)}{N_h(s) + |\mathcal{A}|}$$

$$\leq \frac{H|\mathcal{A}| \log(4|\mathcal{A}|(N + 1)/\delta)}{N_h(s) + |\mathcal{A}|}.$$

Here inequality (a) follows Lemma 3.

When $N_h(s) = 0$, we have that $\pi_h^{\mathrm{BC}}(\cdot|s) = p(\cdot)$ according to Eq.(13). With Lemma 3, we can have that

$$D_{\mathrm{KL}}(\pi_h^{\mathrm{BC}}(\cdot|s), \pi_h^{\mathrm{E}}(\cdot|s)) = D_{\mathrm{KL}}(p(\cdot), \pi_h^{\mathrm{E}}(\cdot|s)) \leq H \log(4|\mathcal{A}|) \leq \frac{H|\mathcal{A}| \log(4|\mathcal{A}|(N + 1)/\delta)}{N_h(s) + |\mathcal{A}|}.$$

By combining the above two cases, we can conclude that with probability at least $1 - \delta$,

$$D_{\mathrm{KL}}(\pi_h^{\mathrm{BC}}(\cdot|s), \pi_h^{\mathrm{E}}(\cdot|s)) \leq \frac{H|\mathcal{A}| \log(4|\mathcal{A}|(N+1)/\delta)}{N_h(s) + |\mathcal{A}|}.$$

Applying the union bound over $(s, h) \in \mathcal{S} \times [H]$ finishes the proof. $\qquad\square$

**Lemma 2** (Sanov's Theorem). *Suppose that $Q$ is a distribution over an alphabet $\mathcal{X}$ and $E$ is a set of distributions over $\mathcal{X}$. Let $\mathcal{D} = \{X_1, X_2, \ldots, X_N\}$ be i.i.d. samples drawn from the distribution $P$. Then*

$$\mathbb{P}(\widehat{P}_{\mathcal{D}} \in E) \leq (N+1)^{|\mathcal{X}|} \exp(-N D_{\mathrm{KL}}(P^\star, Q)),$$

*where $\widehat{P}_{\mathcal{D}}$ denote the empirical distribution from $\mathcal{D}$ and $P^\star = \mathrm{argmin}_{P \in E} D_{\mathrm{KL}}(P, Q)$.*

**Lemma 3.** *For any $(s, h) \in \mathcal{S} \times [H]$, consider $p$ is an uniform distribution over $\mathcal{A}$, we have that*

$$D_{\mathrm{KL}}(p(\cdot), \pi_h^{\mathrm{E}}(\cdot|s)) \leq H \log(4|\mathcal{A}|).$$

*Proof.* For any fixed $(s, h) \in \mathcal{S} \times [H]$,

$$D_{\mathrm{KL}}(p(\cdot), \pi_h^{\mathrm{E}}(\cdot|s)) = \sum_{a \in \mathcal{A}} \frac{1}{|\mathcal{A}|} \log \left( \frac{1/|\mathcal{A}|}{\pi_h^{\mathrm{E}}(a|s)} \right) = -\log(|\mathcal{A}|) - \frac{1}{|\mathcal{A}|} \sum_{a \in \mathcal{A}} \log(\pi_h^{\mathrm{E}}(a|s)).$$

According to Eq.(1), we can further obtain that

$$D_{\mathrm{KL}}(p(\cdot), \pi_h^{\mathrm{E}}(\cdot|s)) = -\log(|\mathcal{A}|) - \frac{\sum_{a \in \mathcal{A}} Q_h^{\star,\mathrm{soft}}(s, a)}{|\mathcal{A}|} + V_h^{\star,\mathrm{soft}}(s).$$

Notice that

$$Q_h^\star(s, a) = \mathbb{E} \left[ \sum_{h'=h}^{H} r_{h'}^\star(s_{h'}, a_{h'}) + \sum_{h'=h+1}^{H} H(\pi_{h'}^{\mathrm{E}}(\cdot|s_{h'})) \,\Big|\, s_h = s, a_h = a, \pi^{\mathrm{E}} \right] \geq 0,$$

$$V_h^\star(s) = \mathbb{E} \left[ \sum_{h'=h}^{H} (r_{h'}^\star(s_{h'}, a_{h'}) + H(\pi_{h'}^{\mathrm{E}}(\cdot|s_{h'}))) \,\Big|\, s_h = s, a_h = a, \pi^{\mathrm{E}} \right]$$

$$\leq (H - h + 1)(1 + \log(|\mathcal{A}|)).$$

Then we have that

$$D_{\mathrm{KL}}(p(\cdot), \pi_h^{\mathrm{E}}(\cdot|s)) \leq (H - h + 1)(1 + \log(|\mathcal{A}|)) \leq H(1 + \log(|\mathcal{A}|)) \leq H \log(4|\mathcal{A}|).$$

$\qquad\square$

**Lemma 4.** *Suppose $n \sim \mathrm{Bin}(N, p)$ where $N \geq 1$ and $p \in [0, 1]$. Then with probability at least $1 - \delta$, we have*

$$\frac{p}{\max\{n, 1\}} \leq \frac{12 \log(2/\delta)}{N}.$$

*Proof.* According to the Chernoff bound (Wainwright, 2019), with probability at least $1 - \delta$,

$$\left| \frac{n}{N} - p \right| \leq \sqrt{\frac{3p \log(2/\delta)}{N}}.$$

This implies a quadratic inequality regarding $x = \sqrt{p}$.

$$x^2 - bx - c \leq 0, \ b = \sqrt{\frac{3 \log(2/\delta)}{N}}, c = \frac{n}{N}.$$

Solving this inequality yields that

$$\sqrt{p} = x \leq \frac{b + \sqrt{b^2 + 4c}}{2} \leq b + \sqrt{c} = \sqrt{\frac{3\log(2/\delta)}{N}} + \sqrt{\frac{n}{N}} \leq 2\sqrt{\frac{3\max\{n,1\}\log(2/\delta)}{N}}.$$

This directly implies that

$$p \leq \frac{12\max\{n,1\}\log(2/\delta)}{N}.$$

Rearanging the above inequality finishes the proof. □

## C. Experiment Details

### C.1. Implementation Details of CoPT-AIL

In this part, we present the detailed implementation of CoPT-AIL, which is outlined in Algorithm 3. In the pretraining stage, we first pretrain the policy via BC.

$$\pi^1 \leftarrow \pi^{\mathrm{BC}}, \ \pi^{\mathrm{BC}} = \underset{\pi \in \Pi}{\mathrm{argmax}} \ \sum_{i=1}^{N} \sum_{h=1}^{H} \log\big(\pi_h(a_h^i \mid s_h^i)\big).$$

Then, according to the analysis in Section 4.1, we pretrain the reward by setting

$$r_h^1(s, a) = \log\big(\pi_h^1(a|s)\big).$$

For continuous action space, we parameterize policies with Gaussian distributions and utilize the log probability of the Gaussian distribution.

After the pretraining phase, we conduct the online AIL process, which alternates between policy and reward updates. In iteration $k$, for the policy update, we first learn the Q-function by minimizing the temporal difference learning objective.

$$\min_{Q \in \mathcal{Q}} \ell^k(Q) := \mathbb{E}_{\tau \sim \mathcal{D}^k}\left[\sum_{h=1}^{H}\Big(Q_h(s_h, a_h) - r_h^k(s_h, a_h) - \overline{Q}_{h+1}^k(s_{h+1}, \pi^k)\Big)^2\right] \tag{14}$$

Here $\mathcal{D}^k$ is the replay buffer consisting of all historical online trajectories and $\overline{Q} = \{\overline{Q}_1, \ldots, \overline{Q}_H\}$ is the delayed target Q-function. Besides, we define that $\overline{Q}_{h+1}^k(s_{h+1}, \pi^k) := \mathbb{E}_{a' \sim \pi_{h+1}^k(\cdot|s_{h+1})}[\overline{Q}_{h+1}(s_{h+1}, a')]$. With the newly learned Q-function $Q^{k+1}$, we update the policy by minimizing the objective of $\ell^k(\pi) := -\mathbb{E}_{\tau \sim \mathcal{D}^k}[\sum_{h=1}^{H} Q_h^{k+1}(s_h, \pi)]$.

For the reward update, the objective function is formulated by

$$\ell^k(r) := \mathbb{E}_{\tau \sim \pi^{k+1}}\left[\sum_{h=1}^{H} r_h(s_h, a_h) + \beta \exp(-r_h(s_h, a_h))\right] - \mathbb{E}_{\tau \sim \mathcal{D}^{\mathrm{E}}}\left[\sum_{h=1}^{H} r_h(s_h, a_h)\right]. \tag{15}$$

Here we add a regularization term $\exp(-r_h(s_h, a_h))$ to improve the stability of reward training, and $\beta > 0$ is the regularization coefficient.

### C.2. Architecture and Training Details

The experiments are conducted on a machine with 64 CPU cores and 4 RTX4090 GPU cores. Each experiment is replicated three times using different random seeds. For each task, we adopt online DrQ-v2 (Yarats et al., 2021) to train an agent with sufficient environment interactions and regard the resultant policy as the expert policy. Specifically, we use 3M environment interactions for `Hopper Hop`, and `Walker Run`, and 1M environment interactions for other tasks. Then we roll out this expert policy to collect expert demonstrations. The number of expert trajectories used for training in each task is provided in 1. The architecture and training details of CoPT-AIL and all baselines are listed below.

---

**Algorithm 3** Practical Implementation of CoPT-AIL

---

**Input:** Demonstrations $\mathcal{D}^{\mathrm{E}}$, replay buffer $\mathcal{D}^1 = \emptyset$.
1: Pre-train a policy via BC based on Eq.(2): $\pi^1 \leftarrow \pi^{\mathrm{BC}}$.
2: Represent a reward through $r_h^1(s, a) = \log(\pi_h^{\mathrm{BC}}(a|s))$.
3: **for** $k = 1, 2, \ldots, K - 1$ **do**
4:     Update the Q-value function by $Q^{k+1} \leftarrow Q^k - \eta_Q \nabla \ell^k(Q)$ from Eq. (14).
5:     Update the policy by $\pi^{k+1} \leftarrow \pi^k - \eta_\pi \nabla \ell^k(\pi)$, where $\ell^k(\pi) := \mathbb{E}_{\tau \sim \mathcal{D}^k}[\sum_{h=1}^H Q_h^{k+1}(s_h, \pi)]$.
6:     Apply $\pi^{k+1}$ to roll out a trajectory $\tau^{k+1}$ and append it to the replay buffer $\mathcal{D}^{k+1} = \mathcal{D}^k \cup \{\tau^{k+1}\}$.
7:     Update the reward function by $r^{k+1} \leftarrow r^k - \eta_r \nabla \ell^k(r)$ from Eq. (15).
8:     Update the target Q-value by $\overline{Q}^{k+1} \leftarrow \tau Q^{k+1} + (1 - \tau)\overline{Q}^k$.
9: **end for**

---

**CoPT-AIL:** Our codebase of CoPT-AIL extends the open-sourced framework of IQLearn. We retain the structure and parameter design of the critic from the original framework, and employ SAC (Haarnoja et al., 2018) with a fixed temperature coefficient for policy update. Note that CoPT-AIL pretrains the reward function using the BC policy. Therefore, we implement the reward model with the same architecture as the actor model. A comprehensive enumeration of the hyperparameters of CoPT-AIL is provided in Table 2.

**BC:** We implement BC based on our codebase. The actor model is trained using Mean Squared Error (MSE) loss over 10k training steps.

**OLLIE:** We use the author's codebase, which is available at https://github.com/HansenHua/OLLIEoffline-to-online-imitation-learning. Note that OLLIE is proposed in the setting of IL with a supplementary dataset. To ensure a fair comparison within our pure online IL setting, we followed the practice recommended in the OLLIE paper to set the supplementary dataset as an empty set. This forces OLLIE to operate using only the provided expert demonstrations, matching our exact data assumptions.

**PPIL:** We use the author's codebase, which is available at https://github.com/lviano/p2il. A comprehensive enumeration of the hyperparameters of PPIL is provided in Table 4.

**IQLearn:** We use the author's codebase, which is available at https://github.com/Div99/IQ-Learn. A comprehensive enumeration of the hyperparameters of IQLearn is provided in Table 5.

**FILTER:** We use the author's codebase, which is available at https://github.com/gkswamy98/fast_irl. A comprehensive enumeration of the hyperparameters of FILTER is provided in Table 6.

**HyPE:** We use the author's codebase, which is available at https://github.com/gkswamy98/hyper. A comprehensive enumeration of the hyperparameters of HyPE is provided in Table 7.

*Table 1.* Number of expert trajectories for each task.

| Task | Expert Trajectories |
|------|---------------------|
| Walker Stand | 10 |
| Walker Run | 20 |
| Walker Walk | 10 |
| Cartpole Swingup | 10 |
| Hopper Hop | 10 |
| Hopper Stand | 10 |
| Finger Spin | 20 |
| Cheetah Run | 20 |

*Table 2.* CoPT-AIL Hyper-parameters.

| Parameter | Value |
|---|---|
| discount factor | 0.99 |
| reward regularization coefficient $\beta$ | 1 |
| temperature coefficient | $10^{-2}$ |
| replay buffer size | $5 \cdot 10^5$ |
| batch size | 256 |
| optimizer | Adam |
| *Reward* | |
| learning rate | $1 \cdot 10^{-5}$ |
| number of hidden layers | 2 |
| number of hidden units per layer | 1024 |
| activation | ReLU |
| *Actor* | |
| learning rate | $3 \cdot 10^{-5}$ |
| number of hidden layers | 2 |
| number of hidden units per layer | 1024 |
| activation | ReLU |
| *Critic* | |
| learning rate | $3 \cdot 10^{-4}$ |
| number of hidden layers | 2 |
| number of hidden units per layer | 256 |
| activation | ReLU |

*Table 3.* OLLIE Hyper-parameters.

| Parameter | Value |
|---|---|
| discount factor | 0.99 |
| reward scale | 5 |
| replay buffer size | $2 \cdot 10^6$ |
| batch size | 256 |
| optimizer | Adam |
| *Reward* | |
| learning rate | $1 \cdot 10^{-5}$ |
| number of hidden layers | 2 |
| number of hidden units per layer | 256 |
| activation | ReLU |
| *Actor* | |
| learning rate | $3 \cdot 10^{-5}$ |
| number of hidden layers | 2 |
| number of hidden units per layer | 256 |
| activation | ReLU |
| *Critic* | |
| learning rate | $3 \cdot 10^{-4}$ |
| number of hidden layers | 3 |
| number of hidden units per layer | 256 |
| activation | ReLU |

*Table 4.* PPIL Hyper-parameters.

| Parameter | Value |
|---|---|
| discount factor | 0.99 |
| gradient penalty coefficient | 10 |
| replay buffer size | $1 \cdot 10^6$ |
| batch size | 256 |
| optimizer | Adam |
| *Actor* | |
| learning rate | $3 \cdot 10^{-4}$ |
| number of hidden layers | 2 |
| number of hidden units per layer | 256 |
| activation | ReLU |
| *Critic* | |
| learning rate | $3 \cdot 10^{-4}$ |
| number of hidden layers | 2 |
| number of hidden units per layer | 256 |
| activation | ReLU |

*Table 5.* IQLearn Hyper-parameters.

| Parameter | Value |
|---|---|
| discount factor | 0.99 |
| gradient penalty coefficient | 10 |
| replay buffer size | $1 \cdot 10^6$ |
| batch size | 256 |
| optimizer | Adam |
| *Reward* | |
| learning rate | $3 \cdot 10^{-4}$ |
| number of hidden layers | 2 |
| number of hidden units per layer | 256 |
| *Actor* | |
| learning rate | $3 \cdot 10^{-4}$ |
| number of hidden layers | 2 |
| number of hidden units per layer | 256 |
| activation | ReLU |
| *Critic* | |
| learning rate | $3 \cdot 10^{-4}$ |
| number of hidden layers | 2 |
| number of hidden units per layer | 256 |
| activation | ReLU |

*Table 6.* FILTER Hyper-parameters.

| Parameter | Value |
|---|---|
| discount factor | 0.98 |
| gradient penalty coefficient | 10 |
| replay buffer size | $1 \cdot 10^6$ |
| batch size | 256 |
| optimizer | Adam |
| *Reward* | |
| learning rate | $8 \cdot 10^{-4}$ |
| batch size | 4096 |
| number of hidden layers | 2 |
| number of hidden units per layer | 256 |
| *Actor* | |
| learning rate | $7.3 \cdot 10^{-4}$ |
| number of hidden layers | 2 |
| number of hidden units per layer | 256 |
| activation | ReLU |
| *Critic* | |
| learning rate | $7.3 \cdot 10^{-4}$ |
| number of hidden layers | 2 |
| number of hidden units per layer | 256 |
| activation | ReLU |

*Table 7.* HyPE Hyper-parameters.

| Parameter | Value |
|---|---|
| discount factor | 0.98 |
| gradient penalty coefficient | 10 |
| replay buffer size | $1 \cdot 10^6$ |
| batch size | 256 |
| optimizer | Adam |
| *Reward* | |
| learning rate | $8 \cdot 10^{-4}$ |
| batch size | 4096 |
| number of hidden layers | 2 |
| number of hidden units per layer | 256 |
| *Actor* | |
| learning rate | $7.3 \cdot 10^{-4}$ |
| number of hidden layers | 2 |
| number of hidden units per layer | 256 |
| activation | ReLU |
| *Critic* | |
| learning rate | $7.3 \cdot 10^{-4}$ |
| number of hidden layers | 2 |
| number of hidden units per layer | 256 |
| activation | ReLU |

## D. Additional Experimental Results

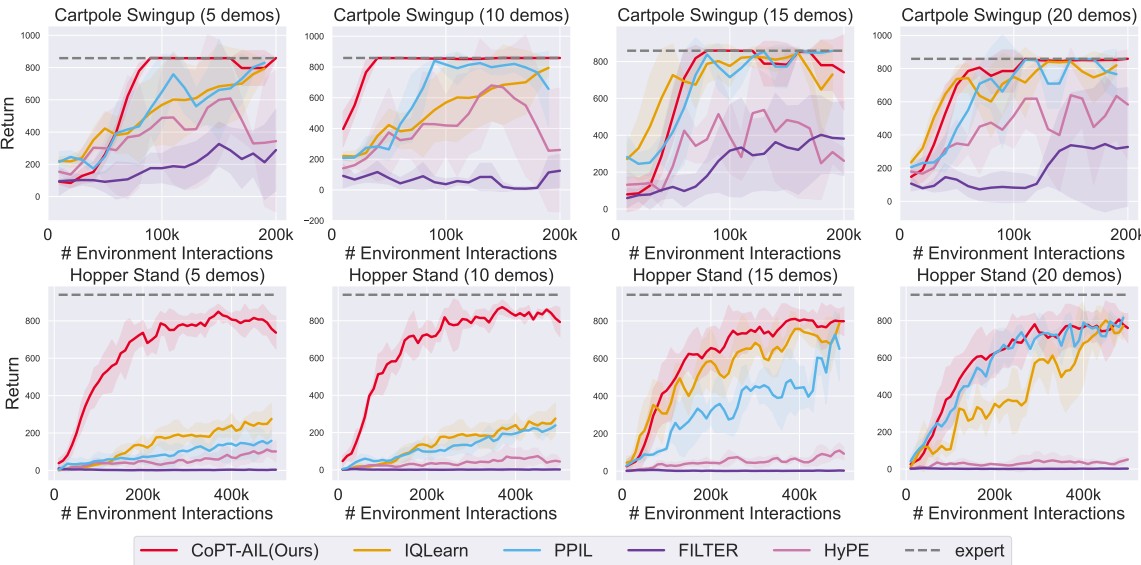

*Figure 4.* Learning curves of CoPT-AIL and the other baselines under different numbers of expert trajectories. Here the $x$-axis is the number of environment interactions and the $y$-axis is the return.

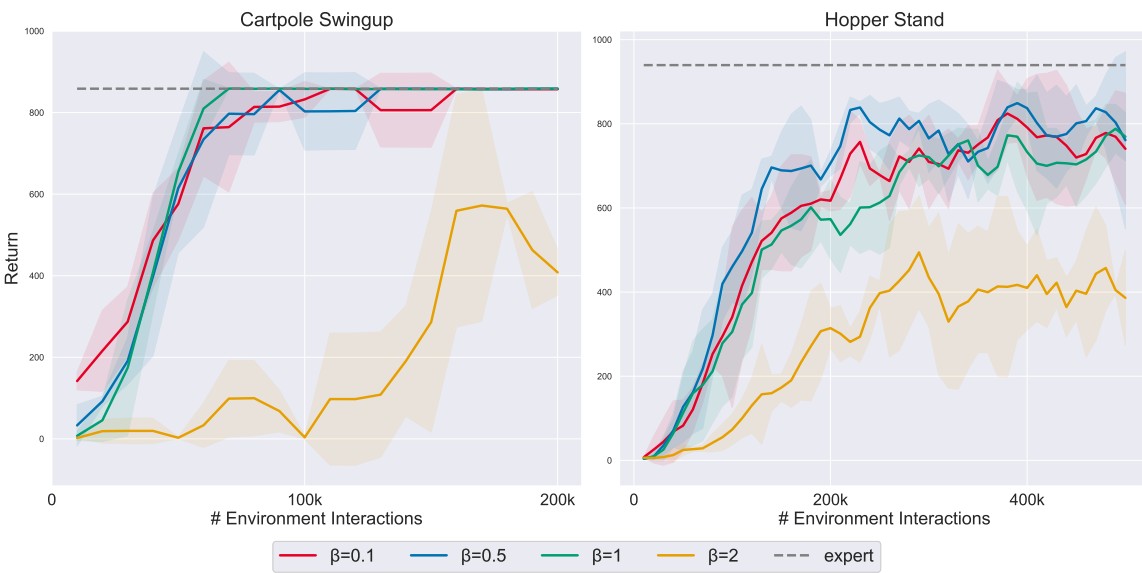

*Figure 5.* Learning curves of CoPT-AIL with different regularization coefficient $\beta$. Here the $x$-axis is the number of environment interactions and the $y$-axis is the return.

### D.1. Sensitivity Analysis

#### D.1.1. SENSITIVITY ANALYSIS ON THE NUMBER OF DEMONSTRATIONS

We conduct a sensitivity analysis on the number of expert trajectories $N$ in $\mathcal{D}^{\mathrm{E}}$. Specifically, we evaluate CoPT-AIL and other baselines with $N \in \{5, 10, 15, 20\}$, and the corresponding learning curves are presented in Figure 4. We observe that CoPT-AIL matches or surpasses the convergence rates of prior SOTA AIL methods across different number of expert trajectories.

#### D.1.2. SENSITIVITY ANALYSIS ON THE REGULARIZATION COEFFICIENT

We also conduct a sensitivity analysis on the regularization coefficient $\beta$ in Eq. (15). Specifically, we evaluate CoPT-AIL with $\beta \in \{0.1, 0.5, 1, 2\}$, and the corresponding learning curves are presented in Figure 5. CoPT-AIL maintains strong performance for $\beta \in [0.1, 1]$. When $\beta$ is large (e.g., $\beta = 2$), performance degrades because the regularization term begins to dominate the reward-training objective and can misguide the reward model.

### D.2. Consistency between Reward Pre-training and Reward Fine-tuning

In CoPT-AIL, reward learning consists of two stages: BC-based reward pre-training and AIL-based reward fine-tuning. We investigate whether any "unlearning" occurs between these stages.

To this end, we first evaluate the gradient cosine similarity between the BC and AIL reward objectives. The resultant curves are shown in Figure 6. We observe that the gradient cosine similarity maintains a high value $\geq 0.8$ (with a maximal possible value of 1) throughout online training. This indicates that the BC and AIL reward objectives are closely aligned rather than working against each other.

Besides, the ultimate goal of reward learning is to produce a reward function that assigns high values to expert data and low values to non-expert data. To verify that the BC and AIL objectives contribute synergistically to this goal, we track the reward gap between expert data and replay buffer data $\mathbb{E}_{(s,a)\sim\mathcal{D}^{\mathrm{E}}}[r_\phi(s,a)] - \mathbb{E}_{(s,a)\sim\mathcal{D}^{\mathrm{replay}}}[r_\phi(s,a)]$ throughout the entire reward learning process. The curves are reported in Figure 7. The first 100K reward gradient steps correspond to the reward pre-training procedure while the remaining gradient steps correspond to the reward fine-tuning procedure. Importantly, when transitioning from pre-training to fine-tuning, the reward gap stays at its previously high level rather than decreasing, indicating that no noticeable unlearning occurs during the switch. This further confirms that the BC and AIL reward objectives operate synergistically, not adversarially.

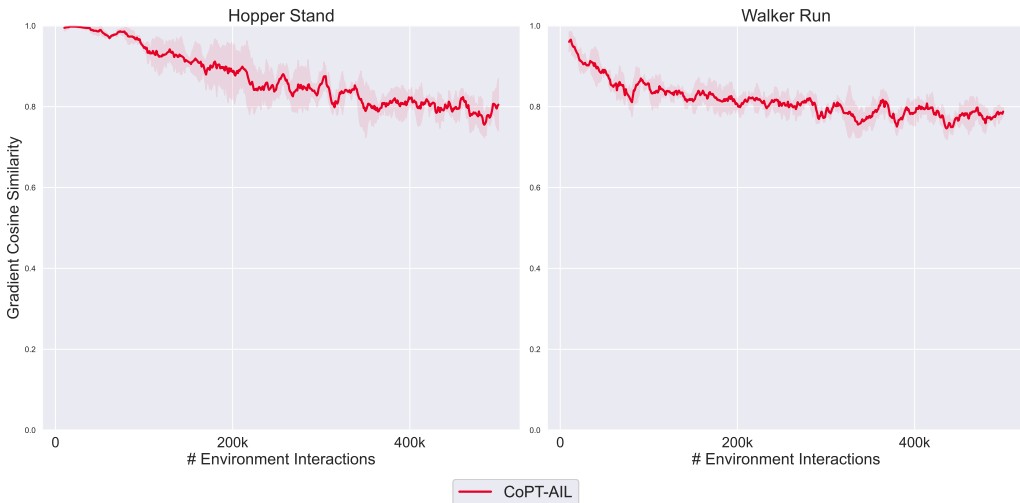

*Figure 6.* Gradient cosine similarity of reward pre-training and fine-tuning objectives in CoPT-AIL. Here the $x$-axis is the number of environment interactions and the $y$-axis is the gradient cosine similarity.

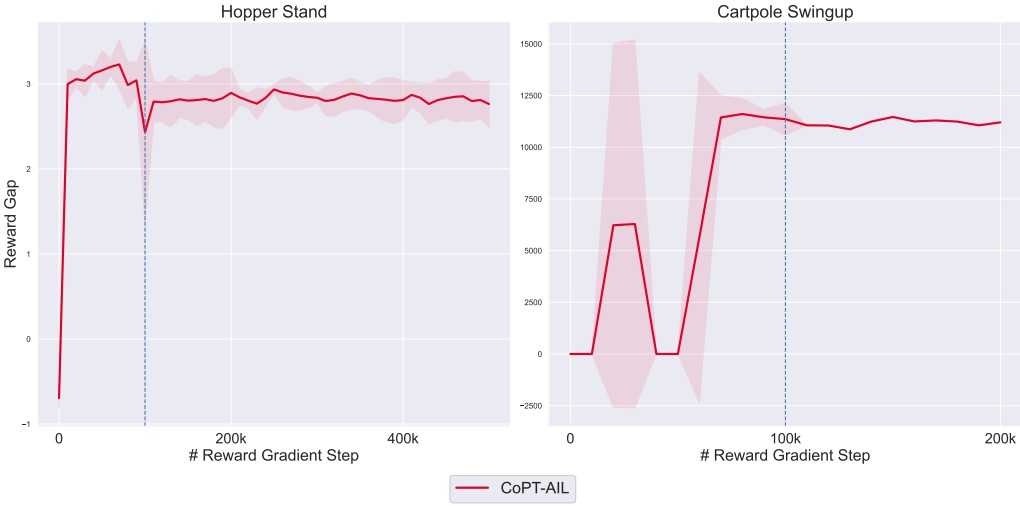

*Figure 7.* Reward gaps between expert data and replay data during the entire reward learning procedure in CoPT-AIL. Here the $x$-axis is the number of reward gradient steps and the $y$-axis is the reward gap $\mathbb{E}_{(s,a)\sim\mathcal{D}^{\mathrm{E}}}[r(s,a)] - \mathbb{E}_{(s,a)\sim\mathcal{D}^{\mathrm{replay}}}[r(s,a)]$. The first 100K reward gradient steps correspond to the reward pre-training procedure while the remaining gradient steps correspond to the reward fine-tuning procedure.

### D.3. Empirical Evaluations of Relative Policy Evaluation Error

The main theoretical prediction from our theory is that the proposed reward pre-training mechanism can reduce the relative policy evaluation error. Here we empirically validate this theoretical prediction by comparing the relative policy evaluation error under the pre-trained reward model and randomly initialized reward model. Recall that the relative policy evaluation error is defined as $(V_{r^\star}^{\pi^E} - V_{r^\star}^{\pi^1}) - (V_r^{\pi^E} - V_r^{\pi^1})$, where $\pi^1$ is the initial policy. Here we use the ground-truth reward defined in DMC tasks as $r^\star$ and approximate the policy value through Monte Carlo estimation using 20 trajectories. Below, we report the normalized relative policy evaluation error divided by the horizon length.

| Task | Walker Stand | Walker Run | Cartpole Swingup | Hopper Hop | Hopper Stand | Finger Spin |
|---|---|---|---|---|---|---|
| CoPT-AIL | 1.51 | 1.49 | -10.77 | 0.63 | 1.84 | 1.93 |
| Baseline | 56.76 | 52.44 | 2.69 | 35.21 | 7.30 | 27.66 |

*Table 8.* Relative policy evaluation error of CoPT-AIL and the baseline across DMControl tasks.

