# OpenReview forum: "Provably Efficient Policy-Reward Co-Pretraining for Adversarial Imitation Learning"
_ICML.cc/2026/Conference — ICML 2026 regular_

### Official Review · Reviewer_wCxq · 2026-02-21

**Soundness:** 4
**Presentation:** 3
**Significance:** 3
**Originality:** 4
**Overall Recommendation:** 4
**Confidence:** 4

**Summary:**

This paper studies how pretraining can accelerate adversarial imitation learning (AIL) and provides theoretical guarantees for pretraining benefits in AIL. The pretraining here refers to training the policy or the reward on the same task before AIL. The authors first analyze AIL with only policy pretraining and decompose the imitation gap into a policy error and a reward error of the first round, showing that the latter dominates when rewards are randomly initialized. Motivated by a reward-shaping analysis and the observation that the log probability of the BC policy can serve as a shaped reward, they propose CoPT-AIL, which jointly initializes the policy via behavior cloning (BC) and the reward as $log(\pi_{BC})$, and they prove improved imitation gap bounds relative to standard AIL. Experiments on 8 DMControl tasks show faster, more stable learning than strong AIL baselines.

**Compliance With Llm Reviewing Policy:**

Affirmed.

**Final Justification:**

I will keep the score.

**Key Questions For Authors:**

- In the experiments, the learning process of CoPR-AIL is more stable than others. How to explain the stability improvement in theory? Moreover, please give a brief description of the baseline methods (the key differences with the proposed methods) to make the paper self-consistent.
- How do the authors set $K$ in the experiment? How do the authors implement argmax and argmin in the experiment? Please provide implementation details.
- How do you reconcile the reward-range assumption r∈[0,1] used to bound Q with the initialization r1(s,a)=log πBC(a|s), which is unbounded below? Do you apply clipping or affine transformations in theory or in practice, and does this affect the shaping invariance or any steps in the proofs?
- Could you include an ablation with reward-only pretraining (r1 from BC, policy initialized randomly) to directly test the “reward error dominates” diagnosis and quantify the marginal benefit of reward initialization versus policy initialization?
- Did the authors attempt using temperature or scaling of $log \pi_{BC}$ as the reward? Did you explore temperature-tuned log-probabilities, clipping, or entropy regularization in the reward initializer, and how do these choices affect stability and sample efficiency?
- What does the shadow area in Figure 2 represent for? Can you report results with additional seeds and confidence intervals?

**Limitations:**

See weakness

**Strengths And Weaknesses:**

Strengths

- This paper provides a novel decomposition of the imitation gap to establish the connection with a reward error term (the quality of the reward on the first round) and a policy error.
- The paper writing is clear and easy to follow.
- In the experiments, the paper compares the proposed method with enough baselines (e.g., FILTER, PPIL, HyPE, IQLearn, OLLIE), indicating consistent or better sample efficiency and stability.

Weaknesses

- In the last line of Algorithm 1, $\bar{\pi}$ is defined as a sample randomly drawn from all the checkpointed policies. How do you implement this in the experiment where the number of these policies $K$ can be very large? This algorithmic design can require a lot of memory, especially when the policy is large. More importantly, how to understand this in terms of theoretical deduction? Does this equivalent to using an average policy of the $K$ policies? I guess not since the authors actually analyze the average performance of the $K$ policies in the deduction of Appendix A.1 Eq. 8. It is this artificial output setting (using a random sample instead of the last checkpoint) that makes the performance depend largely on the performance of the reward function in the first round and then motivates the authors to pretrain the reward function. In the normal setting where only the last checkpoint is evaluated, does the conclusion that the performance depends on the reward function’s quality before the first-round hold?
- Pretraining usually refers to training on a set of diverse tasks instead of just the target task. This paper actually ``warms-up’’ the policy and the reward instead of random initialization. Please consider changing a word or explaining this.
- Deduction of the equation on Line 268 left is needed -- this is not obvious but important to motivate the authors to use the log-probability of the BC policy as the reward. Please also give the deduction in the response for better evaluation of the paper.

---

> ### Author Rebuttal · Authors · 2026-03-31
>
> We appreciate your time to review and provide positive feedback for our work.
>
> **Q1:** The necessity of using a randomly sampled policy as the final output.
>
> **A1:** We respectfully clarify that outputting a uniformly sampled policy is the standard **online-to-batch conversion** technique for minimax optimization [Orabona, 2019].
>
> 1. In theory, AIL is formulated as minimax optimization. Because last-iterate convergence is not guaranteed in minimax optimization, analyzing the average performance is necessary to establish rigorous regret bounds. Thus, the bottleneck caused by the initial reward is a fundamental property of the learning dynamics, not an artificial setting.
> 2. Practically, we evaluate the last checkpoint (standard practice in Deep RL) and our results demonstrate that the theoretical benefits of reward pre-training effectively accelerate this practical convergence.
>
> **Q2:** Consider changing the word "pre-training".
>
> **A2:** We have updated the manuscript to clarify that our method serves as a targeted warm-up phase.
>
> **Q3:** Give the deduction of the equation on Line 268 left.
>
> **A3:** The derivation contains three steps.
> 1. Taking the logarithm of Eq. (1) yields $\log  \pi^{\text{E}}\_{h} (a \mid s) = Q\_{h}^{\star, \text{soft}}(s, a) - V{\_h}^{\star, \text{soft}} (s) $.
> 2. The soft Bellman equation gives $Q\_h^{\star, \text { soft }}(s, a)= r\_h^{\star} (s, a) +\mathbb{E}\_{s^{\prime} \sim P_h(\cdot \mid s, a)} [ V\_{h+1}^{\star, \text { soft }}\left(s^{\prime}\right) ]$.
> 3. Substituting the equation from step 2 into step 1 and rearranging to solve for $r^{\star}_h (s, a)$ yields the exact equation on Line 268.
>
> **Q4:** An explanation for the stability of CoPT-AIL and a description of the baseline methods.
>
> **A4:** **1. Theoretical Explanation for Stability.** CoPT-AIL's stability stems from principled reward anchoring. While standard AIL suffers from chaotic "cold-start" oscillations due to random initialization, our BC-based reward initialization provides a structured, expert-aligned signal from the first iteration. This anchors the adversarial optimization near a meaningful shaping reward, ensuring that subsequent online updates are smooth fine-tuning steps rather than high-variance fluctuations.
>
> **2. Key Differences from Baselines.**
> 1. **Standard AIL (PPIL, FILTER, HyPE)**: These methods optimize from random initializations, making them susceptible to the severe initial reward errors and update instabilities that CoPT-AIL mitigates.
>
> 2. **IQ-Learn.** Although it reformulates AIL as inverse Q-learning, it still leaves the policy and Q-function randomly initialized, lacking the guided warm-start of CoPT-AIL.
>
> 3. **OLLIE.** While OLLIE also uses pre-training, it requires an additional density ratio estimation for its reward warm-up. In contrast, CoPT-AIL elegantly extracts both the policy and reward simultaneously from a single, efficient BC step.
>
> **Q5:** How do the authors set K and implement argmax and argmin in the experiment?
>
> **A5:** We set K=6000 for Hopper Hop&Walker Run tasks and K=2000 elsewhere. Policy argmax and reward argmin updates use SAC and gradient descent, respectively; see Appendix C for details.
>
> **Q6:** How do you reconcile the reward-range assumption [0, 1] with the unbounded reward initialization?
>
> **A6:** Theoretically, we address potential infinite values by adopting an entropy-regularized BC formulation (Tiapkin et al., 2024). This ensures the validity of our error analysis in Theorem 1 while not affecting the shaping invariance established in Proposition 2. In practice, because our Gaussian policies operate within bounded action spaces, the log-probability is constrained. Consequently, no additional clipping or affine transformations are required or used in our implementation.
>
> **Q7:** The reward-only pretraining ablation.
>
> **A7:** We clarify that reducing both policy and reward errors is essential; our "reward error dominates" diagnosis refers to the bottleneck remaining **after policy pre-training is already implemented**. We conducted this ablation and [Fig. R2](https://anonymous.4open.science/r/ICML2026-23985/ablation_R2.pdf) shows that joint policy-reward co-pretraining outperforms both policy-only and reward-only baselines.
>
> **Q8:** Were techniques like temperature scaling used for the BC-based reward initialization?
>
> **A8:** We did not explore these variants, as strictly adhering to our theoretically derived  $\log \pi_{BC}$ initialization already ensures robust performance and stability.
>
> **Q9:** What does the shadow area in Figure 2 represent for?
>
> **A9:** The shadow area in Figures 1 and 2 represents the standard deviation over 3 different seeds.
>
> ---
> We hope that the clarifications on the minimax dynamics and equation derivation resolve your concerns. We respectfully ask you to reconsider your assessment in light of these responses, and we remain open to addressing any further questions.

---

> > ### Author Rebuttal · Reviewer_wCxq · 2026-04-08
> >
> > Thanks the authors for the reply which addresses most of my concerns about the technical details.

---

### Official Review · Reviewer_KLzp · 2026-03-09

**Soundness:** 3
**Presentation:** 1
**Significance:** 2
**Originality:** 2
**Overall Recommendation:** 3
**Confidence:** 4

**Summary:**

This paper is on the high online environment interaction cost issue in adversarial imitation learning (AIL). The authors first conduct a systematic theoretical analysis of AIL with only policy pretraining, decomposing the imitation gap into policy error and reward error. Then, the paper proposes a policy-reward co-pretraining mechanism called CoPT-AIL based on reward-shaping analysis. Experimental results verify that CoPT-AIL converges faster and more stably.

**Compliance With Llm Reviewing Policy:**

Affirmed.

**Key Questions For Authors:**

Please refer to the "Weaknesses" parts. Overall, I think this paper is interesting. If the authors can address my concerns in the rebuttal, I could consider increasing my scores.

**Limitations:**

yes

**Strengths And Weaknesses:**

**Strengths**

1.	Good Theoretical Analysis. The paper makes a pioneering theoretical contribution by providing the first rigorous theoretical analysis of AIL with policy pretraining, clearly decomposing the imitation gap and pinpointing reward error as the core bottleneck.
2.	Novel Algorithm Design. The CoPT-AIL algorithm is designed based on solid reward-shaping theory. To the best of my knowledge, this is novel.
3.	Strong Empirical Validation. The paper compares CoPT-AIL with SOTA AIL methods on 8 DMControl tasks, and the learning curves show that CoPT-AIL has faster convergence speed and better stability.

**Weaknesses**

1.	Limited Theoretical Scope to Tabular MDPs. The theoretical analysis of the paper is limited to the standard tabular MDP setup, which is far from practical scenarios where function approximation is required for high-dimensional state or action spaces.
2.	Lack of Validation on Complex Real-World Tasks. All experiments are conducted on simulated DMControl tasks with relatively simple state representations. We are still unclear whether the co-pretraining mechanism can effectively transfer to these more complex real-world scenarios.
3.	Insufficient Analysis of Long-Horizon Task Limitations. Although the paper mentions that BC policies may suffer from compounding errors in long-horizon rollouts, it does not conduct in-depth analysis of how this affects the performance of CoPT-AIL. There is no experimental comparison on long-horizon tasks.
4.	Lack of other Reward Pretraining Design Space. The paper only uses the log probability of the BC policy as the pretrained reward, without exploring other possible reward pretraining methods (such as using other statistical features of expert trajectories or combining multiple signals).
5.	Improvable Presentation. The presentation of the theoretical part of the paper should be further improved. The current presentation lacks readability.

---

> ### Author Rebuttal · Authors · 2026-03-31
>
> Thanks for the careful review of our paper. We respectfully provide our response as follows.
>
> **Q1:** Limited theoretical scope to tabular MDPs.
>
> **A1**: Our theoretical analysis could be extended to the linear function approximation set-up, where the reward and policy are linear functions of certain features. We would like to point out that the choice of tabular or function approximation set-up primarily influences the analysis for statistical errors arising from finite demonstrations. The main difference is to replace the dependence of $|\mathcal{S}| |\mathcal{A}|$ with the feature dimension $d$. Below we outline how each theoretical result extends to the linear function approximation setting.
>
> 1. **Extension of Proposition 1.**
>    * **Extension of Eq.(4).** Through a concentration argument on the feature space, we can prove that the second term in the RHS of Eq.(4) becomes $\mathcal{O} (H d \sqrt{\frac{\log (dH/\delta)}{N}})$ and the other terms remain unchanged in the function approximation set-up.
>    * **Extension of Eq.(5).** By leveraging the theory of BC under function approximation, we can prove that the last term in the RHS of Eq.(5) becomes $\widetilde{\mathcal{O}} ( \sqrt{\frac{d H^4}{K N}})$ in the function approximation set-up.
> 2. **Extension of Proposition 2.** Proposition 2 holds as stated since its proof is conducted entirely in function space and does not depend on whether the underlying MDP is tabular or continuous.
> 3. **Extension of Theorem 1.** Applying BC guarantees under linear function approximation again yields the upper bound of $\widetilde{\mathcal{O}} (\frac{d H^2}{N})$ for the relative policy evaluation error, which in turn modifies the last term in the RHS of Eq.(7) to $\widetilde{\mathcal{O}} ( \sqrt{\frac{d H^4 }{K N}})$.
>
> **Q2:** Lack of validation on complex real-world tasks.
>
> **A2:** To address this concern, we have conducted additional evaluations on the highly complex, high-dimensional Adroit manipulation benchmark (which was also recommended by Reviewer iabe) and utilized the open-source D4RL datasets (which were also suggested by Reviewer EmJC). **[Fig. R1](https://anonymous.4open.science/r/ICML2026-23985/adroit_R1.pdf) demonstrates that CoPT-AIL consistently matches or exceeds the convergence rates and asymptotic performance of prior SOTA AIL methods.** These results suggest that our co-pretraining mechanism maintains its superior performance on more complex, high-dimensional tasks.
>
> **Q3:** Insufficient analysis of long-horizon task limitations.
>
> **A3:** We provide an analysis of long-horizon task limitations from the following two perspectives.
>
> 1. **Empirical Validation on Long-Horizon Tasks.** The DMControl tasks evaluated in our paper are inherently long-horizon tasks, with a horizon length of 500. Our results explicitly demonstrate that CoPT-AIL successfully overcomes the compounding errors of the initial BC policy through online rollouts. Besides, by translating the optimal offline BC policy into a dense shaping reward, CoPT-AIL effectively accelerates convergence speed compared to both standard AIL and alternative reward pretraining baselines.
> 2. **Sequential Policy Rollouts vs. Local Reward Evaluation.** We emphasize that compounding error is exclusively a problem of the **BC policy**, not the **BC initial reward**. The compounding errors of the BC policy arise from covariate shift during sequential, auto-regressive policy rollouts. However, our shaping reward r(s,a) is evaluated locally on individual state-action pairs. Consequently, using it as an initial reward does not "propagate" rollout-based compounding errors. Instead, it provides a highly accurate, localized, and dense signal that effectively jump-starts the subsequent online AIL process.
>
> **Q4:** Lack of other reward pretraining design space.
>
> **A4**: We appreciate the suggestion, but exploring broader empirical or heuristic reward pre-training designs falls outside the scope of this paper. Our primary objective is to derive a principled reward pre-training method from reward shaping theory, which enables us to establish the first theoretical guarantee for pre-training in AIL. We consider exploring these alternative reward designs in future work.
>
> ---
> We hope that the theoretical extension to function approximation and the new experiments on the Adroit benchmark successfully resolve your concerns. We respectfully ask you to reconsider your assessment in light of these updates and we remain open to any further questions.

---

> > ### Author Rebuttal · Reviewer_KLzp · 2026-04-04
> >
> > For the Q1,Q3 and Q4, although the authors have addressed some of conceptual concerns, the rebuttal is still insufficient to support the paper’s novelty claims. Could the authors give more clarification and evidence?

---

> > > ### Author Response · Authors · 2026-04-06
> > >
> > > We sincerely thank you for reviewing our rebuttal. We would like to respectfully address your new concern regarding the paper's novelty.
> > >
> > > ### 1. Q1, Q3, and Q4 are Addressed and Orthogonal to Novelty
> > >
> > > **We respectfully emphasize that our previous responses have explicitly addressed these specific concerns:** A1 provides the theoretical extension to the function approximation set-up; A3 clarifies that our evaluated benchmarks already consist of long-horizon tasks; and A4 explains that the exploration of heuristic reward designs falls outside the scope of this paper. Crucially, these discussions define the theoretical applicability and empirical scope of this work, which are orthogonal to its novelty.
> > >
> > > ### 2. Core Novelty: Theoretical Analysis and Algorithmic Design for Reward Pre-training in AIL
> > >
> > > We emphasize that the core novelty of our work lies in three principled contributions to reward pre-training in AIL:
> > >
> > > 1. **We develop a theoretical analysis to first diagnose why AIL with policy pre-training alone fails (Section 3).** In Proposition 1, we prove that AIL with BC-pretrained policies suffers from a large reward-error term (called "relative policy evaluation error") caused by randomly initialized rewards. While some empirical works [Sasaki et al., 2018; Jena et al., 2021] observe that policy pretraining can fail, none provide a theoretical explanation. Our work is the first to theoretically diagnose why this failure occurs.
> > >
> > > 2. **We introduce a new policy-reward co-pretraining mechanism based on shaping-reward invariance (Section 4.1).** In Proposition 2, we establish that pretraining toward a shaping reward is sufficient for reducing the "relative policy evaluation error". Combined with the characterization that the log-probability of the expert policy is a shaping reward, we propose a novel policy-reward co-pretraining mechanism. In contrast, prior AIL works do not leverage reward shaping to construct a theoretically grounded reward initialization from policies.
> > > 3. **We provide the first theoretical guarantee for policy-reward co-pretraining improving AIL (Section 4.2).** Theorem 1 demonstrates that CoPT-AIL provably reduces the "relative policy evaluation error", thereby achieving an improved imitation gap bound. To the best of our knowledge, this is the first theoretical guarantee for the efficiency gains of pretraining in AIL. Prior AIL analysis [Shani et al., 2021; Viano et al., 2024] do not provide guarantees for any pre-training strategy.
> > >
> > > ### 3. The Novelty is Widely Recognized across the Review Panel
> > > Crucially, the core novelty of this paper is widely recognized across the review panel.
> > > * Reviewer iabe: Noted it provides "**new insight** into why prior policy-only pretraining fails".
> > > * Reviewer EmJC: Called the central insight "elegant and **novel**" and recognized it as the "**first paper** providing formal theoretical guarantees for the benefits of pretraining in AIL".
> > > * Reviewer KLzp: Highlighted our "**pioneering** theoretical contribution by providing the **first** rigorous theoretical analysis of AIL with policy pretraining" and "**novel** algorithm design".
> > > * Reviewer wCxq: Praised the "**novel** decomposition of the imitation gap".
> > >
> > > ### 4. Summary
> > >
> > > To clearly reflect the current status of our discussion, we summarize how each of your concerns has been addressed in the table below.
> > >
> > > | Reviewer's Concern | Our Response |
> > > | :--- | :--- |
> > > | **Q1: Theoretical Scope to Tabular MDPs**| **Resolved:** In **A1**, we outlined how our theoretical analysis naturally extends to the **linear function approximation** setting . |
> > > | **Q2: Validation on Complex Tasks** | **Resolved:** In **A2**, we conducted substantial new evaluations on the high-dimensional **Adroit manipulation benchmark**. The new results confirm that CoPT-AIL consistently outperforms baselines in complex environments. |
> > > | **Q3: Analysis of Long-Horizon Limitations** | **Clarified:** In **A3**, we clarifed that the evaluated DMControl tasks are **already long-horizon tasks (H=500)**, where CoPT-AIL effectively accelerates convergence speed compared to baselines. |
> > > | **Q4: Other Reward Designs** | **Clarified:** In **A4**, we clarified that the core objective of this paper is to establish a **theoretically principled** reward pre-training framework for AIL. Exploring alternative heuristic reward designs **falls outside the scope of this work**. |
> > > | **Q5 (New): Novelty** | **Clarified:** We emphasized our three unique contributions (theoretical diagnosis, novel co-pretraining design, and the first AIL pre-training guarantee), **which were recognized as novel by the entire review panel**. |
> > >
> > > ---
> > > We sincerely hope that these summarized revisions, alongside our clarifications regarding the paper's core novelty, fully resolve your concerns. We respectfully ask you to reconsider your assessment in light of these comprehensive responses.

---

### Official Review · Reviewer_EmJC · 2026-03-13

**Soundness:** 2
**Presentation:** 2
**Significance:** 2
**Originality:** 3
**Overall Recommendation:** 4
**Confidence:** 4

**Summary:**

This paper provides a theoretical analysis of pretraining in adversarial imitation learning (AIL) and proposes CoPT-AIL, an algorithm that jointly pretrains both the policy and the reward function. The paper analyzes standard AIL with BC-pretrained policies, showing that the imitation gap decomposes into two terms (1) a policy error (KL divergence between expert and initial policy) and (2) a reward error (discrepancy between the true reward and the initial reward). While BC pre-training effectively reduces policy error, the reward error from random initialization persists as a bottleneck.
The key insight is that the log-probability of the expert policy $\log(\pi^{E}(a|s))$ exactly corresponds to a shaping reward of the true reward $r$ in entropy-regularized RL. Further, shaping rewards preserve policy value differences, so initializing the reward as the expert log-probability is sufficient to reduce the relative policy evaluation error without needing to recover true reward itself.
This yields CoPT-AIL, where both policy and reward are derived from a single BC procedure.
With log-probability of BC policy instead of the true expert, Theorem 1 establishes that CoPT-AIL achieves an improved imitation gap bound for standard AIL without pre-training. Experiments on 8 DMControl tasks show CoPT-AIL consistently matches or outperforms prior AIL methods.

**Compliance With Llm Reviewing Policy:**

Affirmed.

**Key Questions For Authors:**

1. The theoretical algorithms (Algorithm 1, 2) use KL-regularized RL, but the practical algorithm (Algorithm 3) uses SAC with entropy regularization. Can the authors justify why Theorem 1's guarantees should be expected to hold for Algorithm 3? Have you considered implementing the KL-regularized policy update (e.g., via PPO/TRPO or an explicit KL penalty) to close this gap?

2. What supplementary offline datasets, if any, were provided to OLLIE in the DMControl experiments? Could you clarify the exact OLLIE implementation used and provide its hyperparameter configurations?

3. Could the authors provide experimental results of CoPT-AIL on a public benchmark such as D4RL? This would enable a direct, apples-to-apples comparison against OLLIE's reported results (e.g., Table 1 in Yue et al., 2024). Since CoPT-AIL operates under a strictly weaker data assumption (expert demonstrations only, no supplementary offline data), even competitive performance on these benchmarks would be a strong argument in CoPT-AIL's favor.

**Limitations:**

yes

**Strengths And Weaknesses:**

## **Strengths:**

1. The central insight of this paper is elegant and novel: the observation that $\log(\pi^{BC}(a|s))$ is a shaping reward of true reward in entropy-regularized RL setting  provides a clean theoretical bridge between reward pretraining and efficiency in IL.  Both policy and reward can be extracted from a single BC procedure, rather than requiring separate pre-training stages, and this is practically appealing.

2. To the best of my knowledge, this is the first paper providing formal theoretical guarantees for the benefits of pretraining in AIL. The theoretical analysis identifying reward error as the bottleneck is a useful diagnostic that could guide future research on pretraining strategies. The practical algorithm is simple to implement (just one extra line: initialize reward as BC log-probability), which increases the likelihood of adoption.

## **Weaknesses:**

1. The paper's framing that "AIL is classified as an online method" (Introduction) is an oversimplification that mischaracterizes the scope of AIL as a field. The online/offline distinction is orthogonal to the AIL objective itself. IQ-Learn (Garg et al., 2021) reformulates the AIL objective as offline soft-Q learning; ValueDICE (Kostrikov et al., ICLR 2020) formulates the dual objective that can be optimized entirely offline. The paper further asserts in Section 2 that "Solving RL problems requires online environment interactions, marking AIL as an online approach," but this premise is incorrect; offline RL methods (CQL, IQL, Decision Transformer, etc.) solve RL problems from pre-collected data without any online interaction. More broadly, the AIL minimax objective (Eq. 3) does not inherently require online interaction; it is the choice of RL method for the inner policy update that determines whether online data is needed. The paper should clarify that it specifically addresses the standard online formulation of AIL (i.e., online policy optimization with online reward updates), rather than implying that AIL is inherently an online method.

2. There is a three-way inconsistency between the expert assumption, the theoretical algorithm, and the practical implementation regarding policy regularization. (a) The expert is assumed to be MaxEnt-optimal (Eq. 1), i.e., the solution to an entropy-regularized RL problem. (b) The theoretical algorithms (Algorithm 1 & 2) use KL-regularized mirror descent with $D_{KL}(\pi \|\pi_k)$ as the regularizer, which enables online regret analysis via Orabona (2019). (c) The practical algorithm (Algorithm 3, Appendix C) uses SAC (Haarnoja et al., 2018) with a fixed temperature coefficient and entropy regularization, a fundamentally different regularizer with a different fixed-point. These are not interchangeable: KL regularization constrains deviation from the previous iterate (trust region), whereas the entropy regularization is relative to the uniform distribution (exploration). Furthermore, since $r^*$ already encodes the expert's entropy-maximizing behavior via the shaped reward $\log \pi^E(a|s)$, optimizing SAC objective does effectively double-counts the entropy term, which the theory does not analyze. Consequently, the theoretical guarantees of Theorem 1 do not formally extend to Algorithm 3.

3. The paper uses self-collected expert demonstrations generated by DrQ-v2 for all DMControl experiments. OLLIE (Yue et al., 2024), which is the most directly comparable and the latest baseline, uses publicly available D4RL datasets (Fu et al., 2020). By not evaluating on these established open-source benchmarks, the paper lacks an opportunity for a direct comparison against OLLIE in its native setting as is.

4. The paper considers OLLIE (Yue et al., 2024) as a baseline, yet (1) the supplementary code contains no OLLIE implementation, and (2) Appendix C provides no hyperparameter table for OLLIE. OLLIE's core mechanism relies on supplementary offline datasets to approximate the GAIL reward function which uses only expert demonstrations. The paper does not describe which supplementary datasets, if any, were provided to OLLIE. If OLLIE was run without its required supplementary datasets, this should be explicitly stated so the reader can interpret the comparison correctly. Without this clarification, the comparison is ambiguous.

---

> ### Author Rebuttal · Authors · 2026-03-31
>
> Thanks for the detailed review of our paper. We respectfully provide our response as follows.
>
> **Q1:** The paper should clarify its specific focus on the standard online formulation of AIL.
>
> **A1:** We will revise the manuscript to explicitly clarify that our theoretical analysis and proposed method specifically target the standard online formulation of AIL.
>
> **Q2:** There is a three-way inconsistency among the MaxEnt expert assumption, the KL-regularized theoretical algorithm, and the entropy-regularized SAC implementation.
>
> **A2:**  We address the reviewer's concerns in two parts:
>
> 1. **There is no inconsistency between assuming a MaxEnt expert and using the KL-regularized objective in Algorithms 1 & 2.** Once the expert policy is provided (regardless of whether it is MaxEnt-optimal), the objective is purely to imitate its behavior. AIL achieves this through state-action distribution matching. This is formalized as the max-min problem of Eq. (3), where the inner RL problem is unregularized. The KL-regularization objective is solely an optimization mechanism (mirror descent) to solve the unregularized RL problem.
>
> 2. **Closing the gap between the KL-regularized theoretical algorithm and the SAC implementation.** To rigorously eliminate this gap and perfectly align our practical implementation with Theorem 1, we have updated CoPT-AIL to use PPO instead of SAC, realizing the KL-regularized mirror descent analyzed in our theory. To validate this theoretically aligned implementation, we conducted new experiments on the high-dimensional manipulation benchmark Adroit (as recommended by Reviewer iabe), utilizing the D4RL expert datasets you suggested. [Fig. R1](https://anonymous.4open.science/r/ICML2026-23985/adroit_R1.pdf) demonstrates that the updated CoPT-AIL consistently matches or exceeds the convergence rates of prior SOTA AIL methods. By resolving the regularization discrepancy and validating the updated algorithm on complex tasks, we successfully bridge the gap between theory and practice.
>
> **Q3:** The paper should evaluate on established open-source benchmarks like the D4RL datasets.
>
> **A3:** We have incorporated the D4RL datasets you recommended. Because D4RL does not include DMControl tasks, we conducted these new evaluations on the high-dimensional Adroit benchmark (also suggested by Reviewer iabe). [Fig.R1](https://anonymous.4open.science/r/ICML2026-23985/adroit_R1.pdf) shows that CoPT-AIL consistently matches or exceeds the convergence rates and performance of prior SOTA AIL methods including OLLIE. This provides the rigorous, apples-to-apples comparison requested on a standardized open-source benchmark.
>
> **Q4:** The paper lacks implementation details for the OLLIE baseline and is ambiguous about whether OLLIE was provided with its required supplementary offline datasets. Evaluating CoPT-AIL on public benchmarks like D4RL would clarify these data assumptions and enable a direct, apples-to-apples comparison.
>
> **A4:** First, regarding the code and hyperparameters: we utilized the official codebase provided by the OLLIE paper and ran the baseline using their officially recommended hyperparameters. We have now added a comprehensive hyperparameter table as follows.
>
> ### OLLIE Hyper-parameters
>
> |Parameter|Value|
> |:---|:---|
> |Discount factor|0.99|
> |Reward scale|5|
> |Replay buffer size|$2 \times 10^6$|
> |Batch size|256|
> |Optimizer|Adam|
> |**Reward:** Learning rate|$1 \times 10^{-5}$|
> |**Reward:** Number of hidden layers|2|
> |**Reward:** Number of hidden units per layer|256|
> |**Reward:** Activation|ReLU|
> |**Actor:** Learning rate|$3 \times 10^{-5}$|
> |**Actor:** Number of hidden layers|2|
> |**Actor:** Number of hidden units per layer|256|
> |**Actor:** Activation|ReLU|
> |**Critic:** Learning rate|$3 \times 10^{-4}$|
> |**Critic:** Number of hidden layers|3|
> |**Critic:** Number of hidden units per layer|256|
> |**Critic:** Activation|ReLU|
>
> Second, regarding the supplementary datasets: to ensure a fair comparison within our pure online IL setting, we followed the practice recommended in the OLLIE paper for scenarios without offline data and set the supplementary dataset to an empty set. This forces OLLIE to operate using only the provided expert demonstrations, matching our exact data assumptions. As detailed in **A3**, we have also added new evaluations using the D4RL datasets on the Adroit benchmark to further solidify this apples-to-apples comparison.
>
> ---
> We hope that **the alignment of our implementation with the theoretical algorithm via PPO** and **the new empirical evaluations on the D4RL datasets** successfully resolve your concerns. We respectfully ask you to reconsider your assessment in light of these updates, and we are open to addressing any further questions you might have.

---

> > ### Author Rebuttal · Reviewer_EmJC · 2026-04-01
> >
> > I thank the authors for their thorough rebuttal. I acknowledge the authors addressed my concerns as follows:
> >
> > - **Q1**: The authors agreed to clarify the online AIL scope.
> > - **Q3**: New experiments on the Adroit benchmark with D4RL expert datasets provide the public-benchmark evaluation I requested. The results are competitive.
> > - **Q4**: The OLLIE implementation details and hyperparameter table are now comprehensive.
> > - **Q2**: The switch from SAC to PPO eliminates the entropy double-counting issue and substantially narrows the gap between the theoretical algorithm and the practical implementation.
> >
> > In **A2**, the authors state that the objective is still valid regardless of whether the expert is MaxEnt-optimal but this claim requires further clarification: the derivation connecting $\log \pi^E$  to the shaping reward relies on Eq. (1), which requires the expert to be the solution of an entropy-regularized MDP. This means Proposition 2, Theorem 1 do not directly extend to settings where expert policies cannot be obtained from the MaxEnt RL objective, such as deterministic experts.
> > Since such settings are common in practical IL, the authors should justify or clarify this point as a limitation in the revised manuscript.
> >
> > Overall, I believe the authors have sufficiently addressed my main concerns, and I am raising my rating accordingly.

---

> > > ### Author Response · Authors · 2026-04-02
> > >
> > > We sincerely thank you for your continued engagement, and we are glad that our rebuttal has sufficiently addressed your main concerns.
> > >
> > > Regarding your follow-up on the maximum entropy (MaxEnt) assumption, we first clarify that our statement in A2 was intended to emphasize that there is no gap between the MaxEnt optimal expert assumption and our Algorithms 1 & 2. We agree with your observation that Algorithm 2 and Theorem 1 do require the expert policy to be MaxEnt optimal.
> > >
> > > While our theory relies on the MaxEnt assumption, this constitutes a broad, foundational, and active area of research. The MaxEnt formulation was originally introduced in [R1] to resolve reward ambiguity in standard inverse reinforcement learning (IRL), and has since become the standard paradigm underlying seminal Deep IRL algorithms [R2, R3]. Recent theoretical works [R4, R5] continue to study reward identifiability within this scope. Our reward pre-training framework thus targets a widely adopted problem setting.
> > >
> > > Nevertheless, your point is exceptionally well-taken. Following your suggestion, we will explicitly add a discussion of this limitation to the "Limitations" section in the revised manuscript, clearly defining the theoretical boundaries of our framework to avoid overclaiming and ensure rigor.
> > >
> > >
> > > We hope this clarification addresses your remaining concern, and we sincerely appreciate your constructive feedback in helping us improve this manuscript.
> > >
> > > **References:**
> > >
> > > [R1] Maximum entropy inverse reinforcement learning. Ziebart et al., 2008.
> > >
> > > [R2] Maximum Entropy Deep Inverse Reinforcement Learning. Wulfmeier et al., 2015.
> > >
> > > [R3] Guided Cost Learning: Deep Inverse Optimal Control via Policy Optimization. Finn et al., 2016.
> > >
> > > [R4] Reward Identification in Inverse Reinforcement Learning. Kim et al., 2021.
> > >
> > > [R5] Invariance in policy optimisation and partial identifiability in reward learning. Skalse et al., 2023.

---

### Official Review · Reviewer_iabe · 2026-03-13

**Soundness:** 3
**Presentation:** 2
**Significance:** 2
**Originality:** 2
**Overall Recommendation:** 4
**Confidence:** 2

**Summary:**

The paper introduces CoPT-AIL, a policy-reward co-pretraining framework for adversarial imitation learning (AIL). It first provides a theoretical diagnosis showing that standard policy pretraining alone (via behavioral cloning) leaves a dominant reward error term unaddressed, then leverages potential-based reward shaping to jointly initialize both the policy and an aligned initial reward using a single BC pass. Theoretical bounds are derived to prove an improved imitation gap compared to vanilla AIL, and experiments on DMControl tasks demonstrate faster convergence with fewer interactions. The work aims to bridge the long-standing theory-practice gap in AIL pretraining.

**Compliance With Llm Reviewing Policy:**

Affirmed.

**Final Justification:**

The authors fully response all my concerns, I'd like to raise my rating.

**Key Questions For Authors:**

1. Given that the paper (and prior work) explicitly shows BC suffers from compounding errors and poor long-horizon performance, why is log(π_BC) still a reliable shaping reward? If BC is “bad,” how does the method avoid propagating those errors into the initial reward signal? A convincing answer or additional analysis (e.g., sensitivity to BC quality) would strengthen my assessment of soundness.
2. The experiments are limited to 8 DMControl tasks. Why not evaluate on the same broader suite used by OLLIE (vision-based Robomimic, Adroit, AntMaze, FrankaKitchen) ? Matching or exceeding OLLIE’s experimental scope would be necessary for me to view the empirical claims as competitive.

**Limitations:**

No. The paper acknowledges the tabular setting limitation in the conclusion but does not discuss (or experiment on) high-dimensional/visual domains or human demonstrations. Adding such experiments would make the limitations section more honest and complete.

**Strengths And Weaknesses:**

Strengths：
1. The theoretical contribution is well-structured: the paper decomposes the imitation gap (Proposition 1), proves that shaping rewards suffice to reduce reward error (Proposition 2), and delivers the formal guarantee that co-pretraining tightens the lower bound of the imitation gap. This provides new insight into why prior policy-only pretraining fails.
2. The algorithm is computationally lightweight: a single BC step simultaneously initializes both policy and reward (r₁ = log π_BC), requiring no extra data or IRL solver, making it easy to plug into existing AIL pipelines.
3. The ablation study cleanly isolates the benefit of joint pretraining which supports the major claim of the paper.

Weaknesses：
1. The experimental validation is insufficient to support the claimed generality. All results are confined to 8 DMControl tasks with synthetic experts; there is no evaluation on vision-based domains, high-dimensional manipulation (Adroit, FrankaKitchen), or real human demonstrations. In contrast, the closely related baseline OLLIE (Yue et al., 2024) demonstrates consistent gains across 20+ tasks including vision and imperfect data, making CoPT-AIL’s empirical claims appear narrow and under-tested.
2. A core inconsistency arises in the method itself: the paper repeatedly shows (and cites) that BC policies suffer from severe compounding errors and perform poorly in long-horizon rollouts, yet it relies on log(π_BC) as the shaping reward initialization. If BC is already “bad,” it is unclear why its log-probabilities should serve as a reliable expert proxy for reward shaping; this circular reliance undermines the practical soundness of the initialization step.

---

> ### Author Rebuttal · Authors · 2026-03-31
>
> Thanks for the careful evaluation of our paper. We respectfully provide our response as follows.
>
> **Q1:** All experimental results are confined to 8 DMControl tasks and do not match the experimental scope of OLLIE.
>
> **A1:** We agree that evaluating on more challenging tasks strengthens our empirical claims. Given the short rebuttal window, we prioritized evaluating CoPT-AIL on the complete Adroit high-dimensional manipulation benchmark (Pen, Hammer, Door, Relocate). **[Fig. R1](https://anonymous.4open.science/r/ICML2026-23985/adroit_R1.pdf) shows that CoPT-AIL consistently outperforms all baselines in both sample efficiency and asymptotic performance in the Adroit benchmark.** These results validate our method's effectiveness well beyond standard DMControl tasks. We are committed to extending our evaluations to other suites (e.g., vision-based tasks).
>
> **Regarding the comparison with OLLIE, we respectfully highlight the fundamentally different scope and nature of the two works.** OLLIE is an empirical paper focused on improving AIL by leveraging supplementary offline datasets. In contrast, as you graciously noted in the "Strengths" section, our submission is primarily a theoretical work operating within the standard online IL setting. Our core contribution lies in formally diagnosing the reward error bottleneck, deriving a principled co-pretraining mechanism, and mathematically proving an improved imitation gap bound.
>
>
> **Because of this fundamental distinction, our experiments are designed primarily as a proof of concept to validate our mathematical predictions, rather than to establish a universal empirical SOTA.** You observed in your review that our ablation study "cleanly isolates the benefit of joint pretraining which supports the major claim of the paper", indicating that our experiments have fulfilled this purpose. We respectfully submit that holding our theoretical analysis to the exhaustive benchmarking standards of an empirical paper like OLLIE creates an asymmetric evaluation. Nevertheless, we hope the newly added Adroit results demonstrate our strong commitment to rigorous empirical validation.
>
> **Q2:** How can $\log (\pi_{\text{BC}} (a|s))$ serve as a reliable shaping reward initialization without propagating compounding errors, given that BC policies are known to perform poorly in long-horizon tasks?
>
> **A2:** The apparent paradox dissolves once we separate two distinct roles: how a policy is executed versus how a reward is evaluated.
>
> We emphasize that compounding error is exclusively a problem of the **BC policy**, not the **BC initial reward**. **Compounding errors in BC arise during sequential, auto-regressive policy rollouts due to covariate shift. However, a shaping reward $r (s, a)$ is evaluated locally on individual state-action pairs.** Because $\pi_{\text{BC}}$ is trained via Maximum Likelihood Estimation, its single-step predictions $\log (\pi_{\text{BC}} (a|s))$ are highly accurate near the expert distribution. Using $\log (\pi_{\text{BC}} (a|s))$ as an initial reward does not "propagate" rollout-based compounding errors; instead, it provides a dense, localized signal that accurately scores state-action pairs near the expert demonstrations. This provides an informative jump-start to the online AIL process, which subsequently uses online interactions to correct compounding errors.
>
> Besides, our reward pretraining operates strictly offline, using only expert demonstrations. In this regime, Rajaraman et al. (2020) rigorously proved that BC is the minimax optimal algorithm. **This establishes a clear theoretical ceiling: without additional online interactions, no algorithm can provably outperform BC.** Therefore, deriving the shaping reward from BC is not a flawed compromise, but the mathematically optimal proxy available during offline pre-training.
>
> ---
> We hope that **the addition of the high-dimensional Adroit benchmark** and **our analytical clarifications regarding the BC-based reward initialization** successfully resolve your concerns. We respectfully ask you to reconsider your assessment in light of these updates, and we are open to addressing any further questions you might have.

---

> > ### Author Rebuttal · Reviewer_iabe · 2026-04-07
> >
> > Fully resolve my concerns, I'd like to raise my rating.

---

### Decision · Program_Chairs · 2026-04-30

**Decision:**

Accept (regular)

**Comment:**

### Summary

This paper theoretically studies the utility of warming up the agent policy and a shaping reward using a behavioral cloning (BC) loss when using adversarial imitation learning (AIL). It follows up this theoretical analysis by presenting an approach called AIL with policy-reward co-pretraining (CoPT-AIL). Evaluations are performed in DMControl with synthetic expert data and in the Adroit domain with D4RL-based expert demonstrations.

### Meta-Review
The theoretical analysis of the warm up phase in AIL was of interest to all the reviewers. There was some pushback against the discrepancy in KL-regularized mirror descent in the theoretical analysis but entropy-regularized training in the empirical evaluation, as well as skepticism whether the DMControl benchmark with synthetic experts are sufficient to evaluate the proposed technique. During the discussion phase these shortcomings seem to have been mitigated for the most part with the use of PPO in the Adroit setting where expert demonstrations are available as part of the D4RL benchmark.
There are some lingering questions about how well the theoretical analysis in the tabular setting translates to more complex domains, and whether there are other reward learning approaches beyond just using the logits of the BC policy as the shaping reward, but these could be construed to be out of scope of the paper as it stands.